# Dynamic High-Resolution Network for Semantic Segmentation in Remote-Sensing Images

**Shichen Guo** [1,2]**, Qi Yang** [2,3]**, Shiming Xiang** [2,3]**, Pengfei Wang** [1] **and Xuezhi Wang** [1,*]

[1]  Computer Network Information Center, Chinese Academy of Sciences, Beijing 100083, China; guoshichen@cnic.cn (S.G.); pfwang@cnic.cn (P.W.)

[2]  University of Chinese Academy of Sciences, Beijing 100049, China; yangqi2021@ia.ac.cn (Q.Y.); smxiang@nlpr.ia.ac.cn (S.X.)

[3]  State Key Laboratory of Multimodal Artificial Intelligence Systems (MAIS), Institute of Automation, Chinese Academy of Sciences, Beijing 100190, China

*  Correspondence: wxz@cnic.cn

**Abstract:**  Semantic segmentation of remote-sensing (RS) images is one of the most fundamental tasks in the understanding of a remote-sensing scene. However, high-resolution RS images contain plentiful detailed information about ground objects, which scatter everywhere spatially and have variable sizes, styles, and visual appearances. Due to the high similarity between classes and diversity within classes, it is challenging to obtain satisfactory and accurate semantic segmentation results. This paper proposes a Dynamic High-Resolution Network (DyHRNet) to solve this problem. Our proposed network takes HRNet as a super-architecture, aiming to leverage the important connections and channels by further investigating the parallel streams at different resolution representations of the original HRNet. The learning task is conducted under the framework of a neural architecture search (NAS) and channel-wise attention module. Specifically, the Accelerated Proximal Gradient (APG) algorithm is introduced to iteratively solve the sparse regularization subproblem from the perspective of neural architecture search. In this way, valuable connections are selected for cross-resolution feature fusion. In addition, a channel-wise attention module is designed to weight the channel contributions for feature aggregation. Finally, DyHRNet fully realizes the dynamic advantages of data adaptability by combining the APG algorithm and channel-wise attention module simultaneously. Compared with nine classical or state-of-the-art models (FCN, UNet, PSPNet, DeepLabV3+, OCRNet, SETR, SegFormer, HRNet+FCN, and HRNet+OCR), DyHRNet has shown high performance on three public challenging RS image datasets (Vaihingen, Potsdam, and LoveDA). Furthermore, the visual segmentation results, the learned structures, the iteration process analysis, and the ablation study all demonstrate the effectiveness of our proposed model.

**Keywords:** semantic segmentation; remote-sensing image; neural architecture search; sparse regularization; HRNet

## 1. Introduction

With the rapid development of remote-sensing (RS) technologies, a large number of RS images are taken by different devices every day. In practice, it is an urgent need to understand well the contents recorded in these images. As a fundamental approach to analyzing RS images in many systems, the task of semantic segmentation is to divide the input image into regions with explicit category labels for downstream tasks. Compared with the task conducted on natural images, segmenting RS images could be more complex. RS images are all taken at a far distance, and there are a lot of confusing objects scattered spatially here and there, with significant variations in size, style, and visual appearance. This causes high intra-class scatter and low inter-class variance in pattern analysis, making it more difficult to achieve satisfactory performance.

Recent years have witnessed the great successes of deep learning in the field of RS image processing. Along this technical line, a fundamental job is to design a good architecture for data adaptability. There are many multi-scale objects for high-resolution RS images with rich ground details. Thus, when designing neural architecture, it is necessary to consider how to achieve multi-scale feature fusion well for fine semantic segmentation. To this end, some thoughtful designs have been demonstrated in the literature [1–7]. The practices have indicated that designing neural architectures that can extract and maintain the representations simultaneously with high, medium, and low resolutions is very important for segmenting the RS objects of different scales well.

To achieve good multi-scale feature fusion, architecture design can be incorporated into the AutoML framework [8–14]. One goal of AutoML is to construct neural architectures automatically by computing itself. Algorithmically, neural architecture search (NAS) methods have been invented to attend to this need. Intrinsically, NAS is an NP-hard problem. Thus, differentiable architecture search algorithms [10,13–16] have gained great attention in recent years due to their relatively low computing complexity. However, there still exist limitations to be overcome for semantic segmentation [17,18]. They tend to a give large chance to the skip connections [19,20] without guaranteeing that the learned architectures have explicit streams to extract the representations with different resolutions.

In the literature, the High-Resolution Network (HRNet) [21,22] is a famous neural architecture for high-resolution representation learning. Technically, it was initially designed for pose estimation [21], and its usage has been demonstrated later in many visual recognition tasks [7,22]. In addition, some variants have been developed, including the higher HRNet [23], the lightweight HRNet [24,25], the dynamic HRNet [26], the HRNet with transformer [27], and so on. Architecturally, the HRNet contains four parallel streams with different resolution representations. It also offers a new mechanism of cross-resolution interaction via dense connections between the streams at different stages. With feature mapping among different resolutions, the HRNet is enabled to capture rich multi-scale information. Therefore, such an architecture could attend well to the needs for the segmentation of RS objects.

Beyond directly applying the primary HRNet [21,22] to RS images, in this study, we start by analyzing the dense connections contained in it. When performing the cross-resolution feature fusion, the HRNet does not consider the contributions of the dense connections and channels, i.e., all of them are equally used. This motivates us to select the important ones by addressing the task in the NAS framework, hoping to enhance its representative capability further. Based on the above observations, in this paper, a Dynamic High-Resolution Network (DyHRNet) is proposed for semantic segmentation in RS images. The DyHRNet is initially constructed on and learned later from the primary HRNet to use its parallel streams with different resolution representations.

The key idea behind the DyHRNet is to evaluate the importance of dense connections and channels for cross-resolution feature fusion. This task is addressed in the NAS framework with channel-wise attention. Mathematically, to avoid solving an NP-hard problem, we choose to relax the 0/1 contributions to be soft ones. With a series of sparse regularizations posed on the learning model, unimportant or useless connections will be identified by assigning low or zero contributions. In this way, the architecture of the primary HRNet is dynamically changed for data adaptability. In addition, a channel-wise attention is designed to evaluate the channel contributions to cross-resolution feature fusion, which further enhances the representation capability of the proposed DyHRNet. Finally, the sparse regularization and the channel-wise attention are combined into a compact optimization model for end-to-end learning. The contributions and the main work are summarized as follows:

- A Dynamic High-Resolution Network (DyHRNet) is proposed for semantic segmentation in RS images. The neural architecture of the DyHRNet is constructed on and learned from the primary HRNet. This task is formulated as a problem of neural

architecture search (NAS) with channel-wise attention. Mathematically, a compact learning model with sparse regularization is developed to achieve this goal.

- Within the Stochastic Gradient Descent (SGD) approach framework for end-to-end training, the sparse regularization subproblem is iteratively solved by the Accelerated Proximal Gradient (APG) algorithm. As a result, the important connections between the parallel streams in the HRNet are selected for cross-resolution feature fusion.
- A mechanism of channel-wise attention is proposed to evaluate the channel contributions for cross-resolution feature aggregation. The attention module has a native structure in which it is easy to format the importance score for channel mapping. As a result, the channel contributions in HRNet are automatically modulated to enhance the representation capability of the DyHRNet.
- The performance of DyHRNet for segmenting RS images has been evaluated on three challenging public benchmarks, including the ISPRS 2D semantic segmentation challenge Vaihingen and Potsdam dataset and the LoveDA dataset. The extensive experiment results with numerical scores and visual segmentation, the learned structures, the iteration process analysis, and the ablation study all demonstrate the effectiveness of the proposed model.

The article is organized as follows: Section 1 describes the background information, the motivation, the objective, and the predictions of this study. Section 2 describes the related works. The details of the proposed method are introduced in Section 3. Experimental results are reported in Section 4. Discussions are given in Section 5, followed by the conclusions in Section 6.

## 2. Related Works

### 2.1. Semantic Segmentation for RS Images

With the great success of deep learning on semantic segmentation for natural images, tremendous efforts have been made by researchers to transfer deep models for RS images [28,29]. Architecturally, most of the models have been formulated with convolution, pooling, and up-sampling operations, such as the Fully Convolutional Network (FCN) [30], the UNet [31], the Pyramid Scene Parsing Network (PSPNet) [32], the DeepLab [33], the OCRNet [34], and so on. Later, some frameworks were constructed on the transformer. In this family, the SEgmentation TRansformer (SETR) [35] and the SegFormer [36] are two famous models. With the usage of encoder–decoder backbones, some variants have been constructed for this issue [1,4,37–40]. For example, the cascaded network with context information fusion was developed to extract confusing artificial objects [1]. The shuffling network is employed to enhance the feature learning ability [38]. These studies have primarily enhanced the semantic segmentation performance for RS images.

Later, more complex models were considered for segmenting RS images. Specifically, Diakogiannis et al. [41] developed an encoder–decoder with multi-tasking inference sequentially on object boundary, segmentation masks, and reconstruction of the input. Zhang et al. [6] employed a high-resolution network with different branches to extract features at both local and global levels. Xu et al. [7] constructed a high-resolution context extraction network to fuse multi-scale contextual information. Liu et al. [5] constructed a new multi-scale U-shaped CNN for extracting buildings in high-resolution RS images, rendering a novel proposal for this issue with multi-task learning to obtain precise masks and help avoid over-fitting. Tang et al. [40] developed a novel self-supervised contrastive learning framework for semantic segmentation in aerial imagery. Within their framework, the distinct characteristic lies in contrastive learning, which is performed both at the feature level and at the semantic level. Furthermore, with the use of the local mutual information that is embedded into the semantic level of contrastive learning, the representation power of the proposed model is largely enhanced for segmentation [40]. In addition, attention modules in different views [42–44] have been designed for fine segmentation. The transformer has recently been employed as the backbone of this task [45,46]. These models

achieve good segmentation in different methods of local, global, and multi-scale feature fusion.

In the literature, there are a few works on the semantic segmentation of RS images under the NAS frameworks. Zhang et al. [47] employed a directed acyclic graph with tricks of Gumbel-max operations under a differentiable searching framework. Later, Wang et al. [48] proposed the decoupling NAS framework with a hierarchical search space for RS objects at the path level, connection level, and cell level. Broni-Bediako et al. [49] developed an evolutionary NAS method for this task. In their framework, gene expression programming, and cellular encoding were employed to represent the encoding scheme for block-building. In summary, although these approaches achieve good performance on accuracy, high computational complexity degrades their real-world applications.

*2.2. Neural Architecture Search*

Recently, constructing neural architectures automatically via NAS has received significant interest in both academia and industry [8–11,13,14]. There are in total three families of NAS methods, namely evolution-based NASs, RL-based NASs, and gradient-based NASs. For example, Ghiasi et al. [11] developed a NAS framework to search for better architectures of a feature pyramid network for object detection. In their work, a novel scalable search space is constructed to cover all cross-scale connections, and a combination of top-down and bottom-up connections is achieved via NAS tricks to fuse multi-scale features [11]. For another example, Weng et al. [12] designed three types of primitive operations on a search space to search U-like backbones for semantic segmentation. In this way, U-like backbone networks can be automatically constructed by stacking the same number of the searched down-sampling cells and up-sampling cells, rendering good performance for semantic segmentation [12]. In the literature, the proposals for NAS within scalable search spaces are rich, demonstrating bright performance enhancements for various types of visual computing tasks.

Documentation about NAS is rich. Here, we only give a brief review of gradient-based NAS methods, which are related to our work in this paper. Since the differentiable architecture search (DARTS) framework was released in 2017 [10], it has become a famous pipeline with gradient-based searching strategies due to their relatively low computational complexities and competitive performance. To reduce the gap between search and evaluation, tricks with progressive differentiable NAS [50], the combination of evaluation and search [51], Gumbel–Softmax [13], and path-level selection [52] have been proposed to achieve the goal of structure generation. However, the DARTS-based algorithms are prone to yield structures much more with skip connections, which limits their power for real-world applications.

As the current task of NAS is to identify a sub-structure or from a previously defined big structure, pruning tricks have been applied to network generation. Earlier, network pruning was performed for model acceleration or compression [53–56]. Recently, sparse representation has also been introduced to this issue. Yang et al. [57] addressed this task as a problem of sparse coding, where differentiable search is achieved within a lower-dimensional space. In addition, Zhang et al. [14] developed a direct sparse optimization to achieve the goal of model pruning. However, network pruning should be strictly performed on a specific architecture, without the ability to generate new topology and operations.

## 3. Method

Our task is to develop a Dynamic HRNet (DyHRNet) for the fine segmentation of RS objects. The task will be formulated as a NAS problem with channel-wise attention. Formally, a compact learning model with sparse regularization is developed to achieve this goal. The details are described in the following subsections.

### 3.1. Problem Formulation

Figure 1 demonstrates the super-architecture constructed according to the rule used in the original HRNet [21,22]. Totally it consists of four parallel streams with different resolution representations, where each row corresponds to a stream of representations with the same resolution. It offers a new mechanism for cross-resolution interaction via dense connections between the streams at different stages. With feature mapping among different resolutions, the HRNet is enabled to capture rich multi-scale information. Therefore, such an architecture could attend well to our needs for RS images.

For clarity, the architecture can be further divided into four stages. The first stage is at the highest resolution. It contains four convolutional layers, which are recorded together by block $\mathbf{O}_{1,1}$ for simplicity. The next three stages cover different streams with high, medium, and low resolutions. More specifically, the second stage contains one group of dense connections, shown as the dash lines between the representations $\mathbf{O}_{1,2}$, $\mathbf{R}_{1,2}$, $\mathbf{O}_{2,2}$, and $\mathbf{R}_{2,2}$ in Figure 1. In addition, there are four and three groups of dense connections, respectively, in the third and fourth stages (Architecturally, one can set any number of groups of dense connections if needed in practice. Without loss of generality, here we take them as those suggested by the original HRNet). Clearly, with these dense connections, multi-scale features are fused. Thus, such an architecture is suitable for segmenting RS images, where objects with different scales locate here and there in the image.

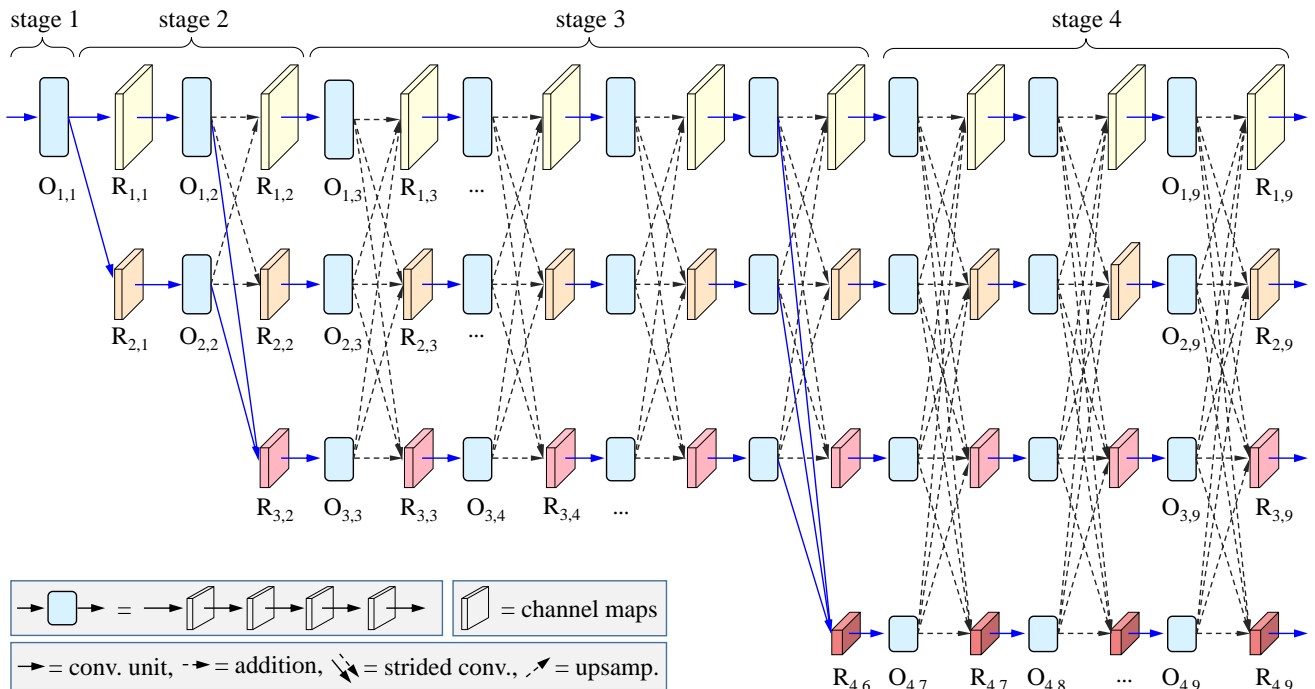

**Figure 1.** The primary HRNet used as a super-architecture to develop the Dynamic HRNet (DyHRNet). Here the connections marked by the dash lines will be selected via sparse optimization.

We denote the representation (namely the output of the four convolutional layers) at the $i$-th row and $j$-th column in Figure 1 by $\mathbf{O}_{i,j}$, and the result of the feature fusion at the same position by $\mathbf{R}_{i,j}$. In the original HRNet, $\mathbf{R}_{i,j}$ is computed as follows:

$$\mathbf{R}_{i,j} = \mathrm{ReLU}\left(\sum_{k=1}^{P} f_{k,j}^{(i)}\left(\mathbf{O}_{k,j}\right)\right),\tag{1}$$

where ReLU stands for the rectified linear unit, $\{f_{k,j}^{(i)}(\cdot)\}$ are the transformation functions and $P$ is the number of streams with different resolutions, which changes within different stages. More specifically, as shown in Figure 1, in the second stage, $i = 1, 2$, $j = 2$ and $P = 2$

; in the third stage, $1 \leq i \leq 3$, $3 \leq j \leq 6$, and $P = 3$; and in the fourth stage, $1 \leq i \leq 4$, $7 \leq j \leq 9$, and $P = 4$. In addition, for the function $f_{k,j}^{(i)}(\cdot)$, in the case of $k = j$, it is an identity mapping; in the case of $k < j$, it is a $3 \times 3$ convolution with stride 2; and in the case of $k > j$, it is a bilinear up-sampling operation with $1 \times 1$ convolution for feature alignment.

As can be seen from Equation (1), the representations in $\{\mathbf{O}_{k,j}\}$ are all equally treated without considering their importance. In other words, there is a lack of a mechanism to evaluate the contributions of the dense connections. Therefore, we cast this task in the NAS framework, which allows us to select those useful connections. Then, Equation (1) is reformulated as follows:

$$\mathbf{R}_{i,j} = \text{ReLU}\left(\sum_{k=1}^{P} s_{k,j}^{(i)}\left(f_{k,j}^{(i)}\left(\mathbf{O}_{k,j}\right)\right)\right), \quad s.t. \quad S_{k,j}^{(i)} \in \{0,1\}, \tag{2}$$

where $\{S_{k,j}^{(i)}\}$ are the selection parameters to be optimized. Technically, in the case of $S_{k,j}^{(i)} = 1$, the candidate link between $\mathbf{O}_{k,j}$ and $\mathbf{R}_{i,j}$ will be selected; and in the case of $S_{k,j}^{(i)} = 0$, the candidate link is useless, and will be discarded. This formulation attends to the task of link search in the NAS work setting. However, it is an NP-hard problem.

To solve this problem, we relax the search space to be a continuous one by allowing each $S_{k,j}^{(i)}$ as a non-negative scaling factor. Then, it turns out that

$$\mathbf{R}_{i,j} = \text{ReLU}\left(\sum_{k=1}^{P} s_{k,j}^{(i)}\left(f_{k,j}^{(i)}\left(\mathbf{O}_{k,j}\right)\right)\right), \quad s.t. \quad S_{k,j}^{(i)} \geq 0, \text{ and } \sum_{k=1}^{P} S_{k,j}^{(i)} < \lambda_j^{(i)}, \tag{3}$$

where $\{S_{k,j}^{(i)}\}$ are the continuous weighting parameters to be learned, the inequality constraint is introduced to force the sparsity of connections, and $\lambda_j^i$ controls the amount of shrinkage for the sparse estimation. That is, a small $\lambda_j$ will force sparser. Algorithmically, the formulation in Equation (3) is a convex relaxing to that in Equation (2).

Please note that the formulation in Equation (3) exhibits another flexible mechanism in that the channel-wise importance can be evaluated jointly. By considering the contributions of channels in each $\mathbf{O}_{k,j}$, it can be rewritten as follows:

$$\mathbf{R}_{i,j} = \text{ReLU}\left(\sum_{k=1}^{P} s_{k,j}^{(i)}\left(f_{k,j}^{(i)}\left(\mathbf{a}_{k,j}^{(i)} \otimes \mathbf{O}_{k,j}\right)\right)\right), \quad s.t. \quad S_{k,j}^{(i)} \geq 0, \text{ and } \sum_{k=1}^{P} S_{k,j}^{(i)} < \lambda_j^{(i)}, \tag{4}$$

where $\mathbf{a}_{k,j}^{(i)}$ is a weighting vector with a length equal to the number of the channels in $\mathbf{O}_{k,j}$, and $\otimes$ stands for the channel-wise product. Technically, channel-wise attention will be designed to fulfill this task (see Section 3.3).

In Equation (4), the weights in $\{s_{k,j}^{(i)}\}$ are learned via the NAS trick, and those in $\{\mathbf{a}_{k,j}^{(i)}\}$ are evaluated via the channel attention. All these weights are positive, which will be modulated by data-driven learning. They may be very small or even zero. In particular, after model training, the cross-resolution connections with zero $s_{k,j}^{(i)}$ and the channels with zero $\mathbf{a}_{k,j}^{(i)}$ will be deleted for prediction.

Now, we can explain the term "dynamic" in our work. In the literature, dynamic models could be developed at different levels of model adaptability in a way of data-driven learning, e.g., at the levels of input data [58], lightweight structures [25], adaptive weights of operations [26], and so on. By contrast, in our work setting, here we first explain its meaning given neural architecture design. The operation in Equation (1) indicates that the neural architecture will remain unchanged before and after training in the original HRNet. With the implementation guided by Equation (4), the connections could be maintained or cut dynamically during and after training. After the model is well trained under the

NAS framework, the connections with $S_{k,j}^{(i)} = 0$ will be deleted from the original structure, yielding a new architecture for segmenting RS images. Then, we explain its meaning given channel-wise importance. The operator "$\otimes$" in Equation (4) will be performed channel by channel with different weights. In this way, the dynamic merit will be demonstrated in the use of channel-wise importance, which will be learned to modulate its contribution. As a whole, the NAS trick and the channel-wise attention will be combined via Equation (4) to develop a dynamic HRNet.

The structure in Figure 1 will be employed as a candidate backbone for architecture optimization. Thus, it is necessary to contain a head module to output semantic segmentation results. Accordingly, our DyHRNet has four groups of parameters to be optimized. The first group collects the parameters in all of the convolution kernels in Figure 1, which are recorded together by variable **W**. The second group consists of those in the channel-wise attention module used to estimate $\{\mathbf{a}_{k,j}^{(i)}\}$, which are collected into variable **U**. The third group includes those in the head part (namely the decoder module), which are collected into variable **H**. The fourth group collects all $\{S_{k,j}^{(i)}\}$ in Equation (3) , which are collected orderly into vector **s**. Then, we have the following optimization problem for segmenting RS objects:

$$
\begin{aligned}
\min_{\mathbf{s}} \quad & L_{validation}(DyHRNet(\mathbf{W}^*, \mathbf{U}^*, \mathbf{H}^*, \mathbf{s})) \\
s.t. \quad & \sum_{(k,j,i)\in\mathcal{N}} S_{k,j}^{(i)} \leq \lambda, \\
& S_{k,j}^{(i)} \geq 0, \\
& (\mathbf{W}^*, \mathbf{U}^*, \mathbf{H}^*) = \arg\min_{\mathbf{W},\mathbf{U},\mathbf{H}} L_{train}(DyHRNet(\mathbf{W}, \mathbf{U}, \mathbf{H}, \mathbf{s})),
\end{aligned}
\tag{5}
$$

where $L_{train}(\cdot)$ is the loss calculated on the training samples, $\lambda$ shrinks together all the controlling factors $\lambda_j^{(i)}$ in Equation (4), and set $\mathcal{N}$ collects the triples $\{(i,j,k)\}$ according to the dash-line links in Figure 1.

According to the relationship between the shrinkage constraints and the regularization representation for $\lambda$ used in LASSO [59], Problem (5) can be reformulated as follows:

$$
\begin{aligned}
\min_{\mathbf{s}} \quad & L_{validation}(DyHRNet(\mathbf{W}^*, \mathbf{U}^*, \mathbf{H}^*, \mathbf{s})) + \lambda\|\mathbf{s}\|_1, \\
s.t. \quad & (\mathbf{W}^*, \mathbf{U}^*, \mathbf{H}^*) = \arg\min_{\mathbf{W},\mathbf{U},\mathbf{H}} L_{train}(DyHRNet(\mathbf{W}, \mathbf{U}, \mathbf{H}, \mathbf{s})),
\end{aligned}
\tag{6}
$$

where $\|\mathbf{s}\|_1$ denotes the $L_1$ norm of **s**.

In Problem (6), there are two subtasks that should be solved iteratively. One is to optimize **W**, **U** and **H**, given **s**; and another is to optimize **s**, given **W**, **U** and **H**. The former can be learned by the algorithm of back propagation of gradients. The latter is difficult to deal with because of the term of $\|\mathbf{s}\|_1$. In the following subsections, we will describe how to solve **s** and how to design the channel-wise attention module to calculate $\{\mathbf{a}_{k,j}^{(i)}\}$ in Equation (4)).

### 3.2. Solving the Sparse Regularization Subproblem with Accelerated Proximal Gradient Algorithm

In this subsection, we use one of the dense connection units at stage 4 of HRNet to illustrate the whole sparse optimization process in the order from Figure 2a–c. To be more specific, Figure 2a depicts the dense cross-resolution connections of the original HRNet, which are also the candidates to be selected by the APG algorithm. At the beginning of the training stage, the weights of these connections are all set to 1.0 to guarantee that all of them have an equal probability to be selected. Figure 2b visualizes a group of learned weights using a solid line and a dashed line. The thicker the solid line is, the greater the weight is and the more important the connection is. In particular, the dashed lines indicate those connections are of zero importance, which could be directly cut off. Figure 2c shows the finally selected connections, which will be used at the inference stage of DyHRNet.

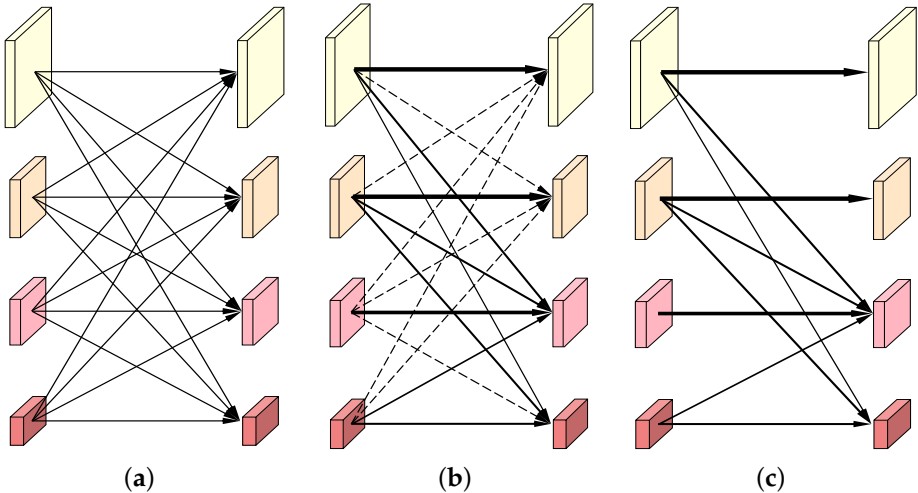

**Figure 2.** The process of sparse optimization is described by taking one group of the dense connections as an example in the order from (**a**) to (**b**) and (**c**). (**a**) The dense connections; (**b**) The weighted and pruned connections; (**c**) The final connections.

Unfortunately, solving the variable **s** in Problem (6) is a challenging task due to the sparse regularization term. One natural selection is to employ the traditional LASSO algorithm [59] to solve it. However, it is uneasy to unfold the mapping function $DyHRNet(\cdot)$ for deduction since it is a hierarchically composite function along the architecture of the DyHRNet. In addition, this could be more difficult since the loss function is defined on all the training samples, and the learning is data-driven in the stochastic work setting.

Alternatively, we employ the Accelerated Proximal Gradient (APG) algorithm [14,60] to optimize **s**. This algorithm has a theoretically sound foundation defined by the proximal algorithms. For convenience, a new function $f(\mathbf{s})$ is introduced to denote the objective function in Problem (6):

$$f(\mathbf{s}) = g(\mathbf{s}) + \lambda \|\mathbf{s}\|_1, \tag{7}$$

where

$$g(\mathbf{s}) = L_{validation}(DyHRNet(\mathbf{W}^*, \mathbf{A}^*, \mathbf{H}^*, \mathbf{s})). \tag{8}$$

Please note that, based on Equation (4), $g(\mathbf{s})$ is a differentiable function with respect to **s**. Now, we make a quadratic approximation to $g(\mathbf{s})$ around current **s**. Then, it follows that

$$g_v(\mathbf{z}) = g(\mathbf{s}) + \nabla g(\mathbf{s})^T (\mathbf{z} - \mathbf{s}) + \frac{1}{2v} \|\mathbf{z} - \mathbf{s}\|_2^2, \tag{9}$$

where $\nabla g(\mathbf{s})$ is the gradient vector of function $g(\cdot)$ at **s**, and $v$ is a positive factor. Thus, based on Equation (9), the task of minimizing $f(\mathbf{s})$ is now updated as

$$\min_{\mathbf{z}} \quad g_v(\mathbf{z}) + \lambda \|\mathbf{z}\|_1. \tag{10}$$

Equivalently, the optimum to Problem (10) can be obtained via the following problem:

$$\text{prox}_{v,\lambda}(\mathbf{y}) = \arg \min_{\mathbf{z}} \quad \frac{1}{2v} \|\mathbf{z} - \mathbf{y}\|_2^2 + \lambda \|\mathbf{z}\|_1, \tag{11}$$

where $\mathbf{y} = \mathbf{s} - v\nabla g(\mathbf{s})$, which is known at current iteration. In addition, here "prox" is known as the proximal operator [60] and gives the optimum to Problem (10).

By further introducing the soft-thresholding operator [60], it turns out that

$$\left[\text{prox}_{v,\lambda}(\mathbf{y})\right]_m = \begin{cases} y_m - \lambda, & y_m \geq \lambda, \\ 0, & |y_m| < \lambda, \\ y_m + \lambda, & y_m \leq -\lambda, \end{cases} \tag{12}$$

where $[\cdot]_m$ stands for the $m$-th entity of the vector and $y_m$ is the $m$-th entity of **y**.

Now the original function $f(\mathbf{s})$ in Equation (7) can be minimized iteratively. With a momentum term to obtain a smooth solution path, we have

$$\mathbf{s}^{(t)} = \mathrm{prox}_{v,\lambda}\left(\mathbf{y}^{(t)} - \eta_t \nabla g(\mathbf{y}^{(t)})\right), \tag{13}$$

in which

$$\mathbf{y}^{(t)} = \mathbf{s}^{(t-1)} + v_t(\mathbf{s}^{(t-1)} - \mathbf{s}^{(t-2)}), \tag{14}$$

where the superscript $(t)$ indicates the $t$-th iteration, $\eta_t$ is a learning ratio and $v_t$ is a contribution factor for historical solution. According to the suggestion given in [60], $v_t$ can be taken as $t/(t+3)$. As the number of iterations increases, it tends to be 1. Thus, $v_t$ is fixed as 0.9 during iteration in our work.

In the first two iterations, both $\mathbf{s}^{(0)}$ and $\mathbf{s}^{(1)}$ are set to be a vector with all entities equal to 1. This means that all the connections in Figure 1 will be initially considered. When **s** is iteratively solved, all the connections will be assigned different weights to indicate their contributions to the final task.

### 3.3. Channel-Wise Attention for Feature Aggregation

As mentioned in Section 3.1, we introduce a channel-wise weighting operation in Equation (4) to develop the mechanism of the dynamic channel and enhance the flexibility of feature aggregation. Intrinsically, this can be addressed as an attention mechanism, which has been widely used in deep neural networks [61,62]. A similar idea has also been applied to the dynamic lightweight HRNet for pose estimation [25]. In this way, channel-wise attention can give larger weights to those important channels and lower weights to those unnecessary ones.

The main task here is to construct the modules to estimate the weighting vectors $\{\mathbf{a}_{k,j}^{(i)}\}$ in Equation (4). Motivated by the kernel aggregation used in [25], our module will be constructed on the representations $\mathbf{O}_{k,j}$ for dense links.

Without loss of generality, we take one group of dense connections in the fourth stage in Figure 1 as an example to explain how to design the attention module. Figure 3 illustrates the detailed layers. Given the representation $\mathbf{O}_{k,j}$, the channel features will be extracted by the Global Averaged Pooling (GAP). In this way, $\mathbf{O}_{k,j}$ will be transformed from a tensor to be a vector with a length equal to the number of the channels in $\mathbf{O}_{k,j}$. Then, it is pushed into the first Fully Connected (FC) layer, followed by the ReLU operation and the second FC layer. The final weighting vector with a length equal to the number of the channels in $\mathbf{O}_{k,j}$ will be output by the Sigmoid layer. Formally, we have

$$\mathbf{a}_{k,j}^{(i)} = \sigma^{(i)}\left(\mathrm{FC}\left(\mathrm{ReLU}\left(\mathrm{FC}\left(\mathrm{GAP}\left(\mathbf{O}_{k,j}\right)\right)\right)\right)\right), \quad i = 1,2; \text{ or } 1 \leq i \leq 3; \text{ or } 1 \leq i \leq 4, \tag{15}$$

where $\sigma(\cdot)^{(i)}$ stands for the final layer with Sigmoid as its activation function.

Finally, it is worth pointing out that the above module "GAP-FC-ReLU-FC-Sigmoid" will be re-used a few times. As illustrated in Figure 3, it will be copied four times to calculate four weighting vectors $\{\mathbf{a}_{k,j}^{(i)}\}$ with $1 \leq i \leq 4$, all taking $\mathbf{O}_{k,j}$ as their input. In this way, the channel importance of the feature maps is considered with adequate nonlinearity for semantic segmentation.

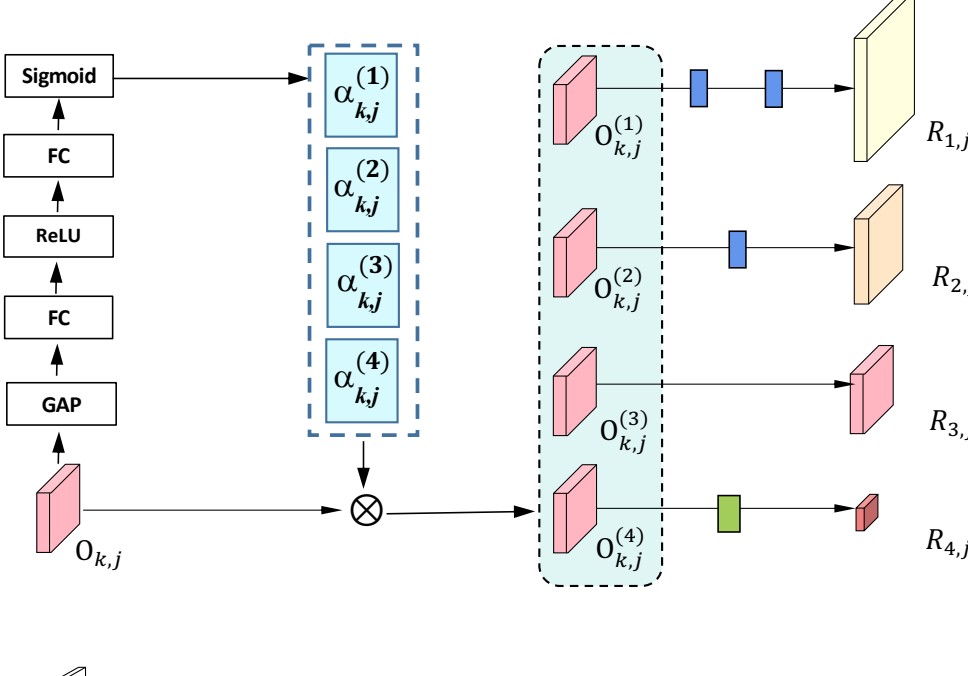

$\square$ = channel maps, $\blacksquare$ =strided 3×3, $\blacksquare$ = upsamp. 1×1

**Figure 3.** As example design of the channel-wise attention module at the fourth stage in Figure 1. Each module is comprised of five layers of "GAP-FC-ReLU-FC-Sigmoid". In this example, it will be copied four times with the same input to calculate $\mathbf{a}_{k,j}^{(i)}$, $1 \leq i \leq 4$.

### 3.4. The DyHRNet Neural Architecture

Based on the descriptions in Sections 3.2 and 3.3, we can now combine them to develop our DyHRNet for semantic segmentation of RS images. Please note that its backbone is optimized from the super-architecture in Figure 1 by applying the APG algorithm described in Section 3.2. It outputs the four representations $\mathbf{R}_{1,9}$, $\mathbf{R}_{2,9}$, $\mathbf{R}_{3,9}$ and $\mathbf{R}_{4,9}$ with different sizes. Accordingly, the the latter three representations will be bilinearly up-sampled, respectively, to be one with size equal to $\mathbf{R}_{1,9}$. Then, they are concatenated together and further transformed by a $1 \times 1$ convolution operation. Finally, we employ the object-contextual representation (OCR) scheme [34] (In the literature, there are many existing modules that can fulfill this task. Based on the empirical observations in [22], we follow the proposal to take the OCR scheme as the head part in our network.) as the decoder to format the output for semantic segmentation.

For clarity, Figure 4 demonstrates the overview of our DyHRNet. It consists of three parts. The first part is the encoder learned from the super-architecture, as demonstrated in Figure 1, which is responsible for feature extraction from the input images. The second part is just a concatenation unit followed by a $1 \times 1$ convolution operation. This treatment achieves multilevel feature fusion for decoding. The third part is the head sub-module of the OCR network [34] employed as the decoder to filter out the abstract features for semantic segmentation. As a whole, these three parts are combined as a whole dynamic network for end-to-end training.

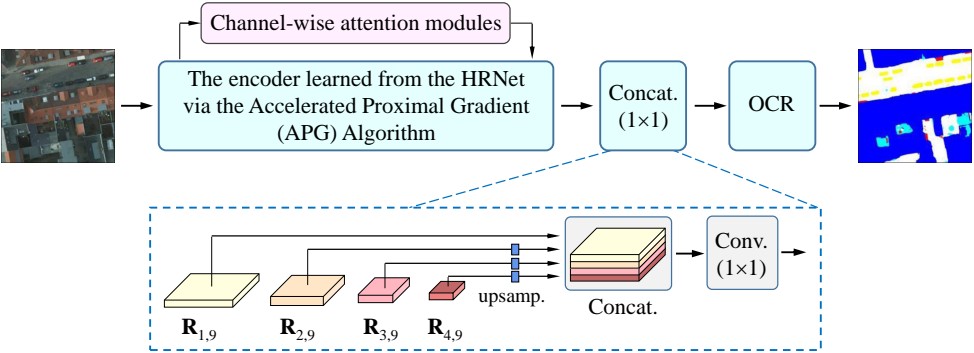

**Figure 4.** Overview of the neural architecture of DyHRNet.

### 3.5. Training and Inference

The original RS images and their ground-truth segmentations are taken to train the model in the learning stage. For each image $\mathcal{I}$ and its ground-truth $\mathcal{Y}$, we denote the predicted segmentation by $\hat{\mathcal{Y}}$. The loss function in Problem (6) is defined as follows:

$$L_{loss}(DyHRNet(\boldsymbol{\theta}, \mathbf{s})) = -\frac{1}{w \times h \times n} \sum_{\mathcal{I} \in trainset} \sum_{z_i \in I} \sum_{k=1}^{C} \delta(y_i = k) \log p_k(z_i), \quad (16)$$

where $\boldsymbol{\theta}$ collects all of the parameters in $\mathbf{W}$, $\mathbf{U}$ and $\mathbf{H}$, "trainset" indicates the training subset, $C$ is the number of categories, $\delta(\cdot)$ is the truth function, $z_i$ is the $i$-th pixel in image $\mathcal{I}$, $y_i$ is the ground-truth label of $z_i$ and $p_k(z_i)$ is the output probability at the $k$-th channel for pixel $z_i$, $w$ and $h$ are the width and height of the training images, and $n$ is the total number of the images in the training set.

In Problem (6), there are two sub-problems to be solved. Technically, we solve them iteratively by fixing $\boldsymbol{\theta}$ or $\mathbf{s}$ once a time for another. Algorithm 1 lists the steps of how to train the DyHRNet. The learning rate $\eta$ takes for gradient update when using the stochastic gradient descent (SGD) strategy to train the model. Except for the OCR module, there are in total more than 170 convolution operations in the DyHRNet. Batch normalization is performed after each convolution operation to guarantee convergence.

---

**Algorithm 1:** Training algorithm for the proposed DyHRNet.

**Input:** RS images with ground-truth segmentations, regularization parameters $\lambda$, learning rate $\eta$, and maximum number of iterations $T$.

**Output:** Parameter $\boldsymbol{\theta} = (\mathbf{W}, \mathbf{U}, \mathbf{H})$ and parameter $\mathbf{s}$.

1   Initialize $\boldsymbol{\theta}$ and $\mathbf{s}$ .
2   Train $\boldsymbol{\theta}$ using SGD for several epochs with mini-batches and batch normalization by fixing $\mathbf{s}$, and obtain $\boldsymbol{\theta}^{(0)}$.
3   Let $t \leftarrow 0$.
4   **while** $t < T$ **do**
5      Fix $\boldsymbol{\theta}^{(t)}$, update parameter $\mathbf{s}$ iteratively using the APG algorithm in Section 3.2, with Equations (12)–(14), and obtain $\mathbf{s}^{(t)}$.
6      Fix $\mathbf{s}^{(t)}$, update parameter $\boldsymbol{\theta}$ iteratively using SGD for several epochs with mini-batches and batch normalization, and obtain $\boldsymbol{\theta}^{(t)}$.
7      **if** $L_{loss}(DyHRNet(\boldsymbol{\theta}, \mathbf{s}))$ *converges* **then**
8         Stop
9      **end**
10     $t \leftarrow t + 1$.
11 **end**

---

When performing the convergence check in Step 8 in Algorithm 1, the convergence condition is that the loss of the network maintains unchanged at two adjacent iterations. After the model is trained, it can be used for RS images with sizes larger than the training images. In this case, the image will be divided into several overlapped patches for segmentation, where the class probabilities of the pixels in the overlapped regions will be averaged to make the final inference.

## 4. Experiments

### 4.1. Data Description

The performance of the proposed DyNRNet has been evaluated on three public challenging benchmark datasets. The details of the datasets are described as follows:

**Vaihingen**: The Vaihingen dataset includes 33 images collected by an aerial camera. The size of the images is about 2494 × 2064 pixels on average, and the Ground Sampling Distance (GSD) or the spatial resolution on the ground is 9 cm. This dataset was constructed in a relatively small village with many buildings and roads. Each sample contains three images with true orthophoto (TOP), digital surface model (DSM), and ground truth. In this dataset, the TOP is composed of red and green bands. Ground truth contains six categories: impervious surface, building, low vegetation, tree, car, and cluster/background. In our experimental setup, only the TOP and ground truth of each sample were used without the DSM. A total of 344 samples are randomly obtained from 15 images for training, and 398 samples are randomly picked out from the remaining 18 images for testing. The samples are all cropped into patches of 512 × 512 pixels.

**Potsdam**: The Potsdam dataset has 38 samples with fine spatial resolution. On average, the size of the images is about 6000 × 6000 pixels, and the GSD is 5 cm. The samples in this dataset were taken from a scene in Potsdam City. Each sample contains three images with TOP, DSM, and ground truth, respectively. Each TOP includes the red, green, and blue bands. It has the same categories as those in the Vaihingen dataset. The DSM is not used for learning. A total of 3456 samples are randomly obtained from 24 images for training, and 2016 samples are randomly picked out from the remaining 14 images for testing. The samples are all cropped into patches of 512 × 512 pixels.

**LoveDA**: The LoveDA dataset contains 5987 samples [63]. Each image has 1024 × 1024 pixels, and the GSD is 0.3 m. The images in this dataset are collected in two senses: urban and rural. Each image includes red, green, and blue bands. The ground truth contains seven categories: building, road, water, barren, forest, agriculture and background. A total of 2522 images are taken for training, and 1669 images are used for testing.

In the phase of model training, data augmentation tricks are employed to enlarge the training samples, including random horizontal flipping, random vertical flipping, and random scaling from a range in {0.5, 0.75, 1.0, 1.25, 1.5} (all with equal probability). In addition, the brightness and contrast of the input image are also randomly changed for training. Then, patches with 512 × 512 pixels are cropped from these images. They are finally organized as a training and testing dataset for model training, evaluation, and comparison.

### 4.2. Compared Models and Experiment Settings

The proposed DyHRNet was compared with the nine classic or state-of-the-art (SOTA) deep learning models for semantic segmentation. These models achieve multi-scale feature fusion for fine segmentation in different ways. For convenience, we summarize them as follows:

- **FCN**: It is a seminal work for semantic segmentation [30]. Currently, it is usually taken as a baseline for comparison. In our experiments, we use the ResNet-101 [64] as its encoder.
- **UNet**: This model contains a contracting path and a symmetric expanding path for multi-scale feature fusion [31]. It is initially designed for biomedical image seg-

mentation, and later widely applied to other types of images. Here, the backbone UNet-S5-D16 is taken as its encoder.

- **PSPNet**: It consists of a pyramid parsing module for global prior representation [32]. The concatenation of multi-scale pyramid representations is transformed to obtain the final per-pixel prediction. The ResNet-101 is employed as its encoder.
- **DeepLabV3+**: This model has an encoder–decoder structure [33]. In DeepLabV3+, the Xception model is modified to extract dense feature maps, and the depthwise separable convolution is employed to design the atrous spatial pyramid pooling and decoder modules. In the experiments, the ResNet-101 is employed as its encoder.
- **OCRNet**: This model fulfills the task of semantic segmentation via three main steps [34]. First, the contextual pixels are divided into a set of soft object regions. Second, the representations of the pixels in each object region are aggregated to obtain object-level representation. Finally, the representation of each pixel is augmented by object-contextual representation (OCR). The ResNet-101 is taken as its encoder.
- **SETR**: The SEgmentation TRansformer (SETR) is a SOTA model [35]. It is a pure transformer. Each image is encoded as a sequence of patches. The VIT-L is employed as its encoder.
- **SegFormer**: It is also a SOTA model constructed on transformers [36]. It has a hierarchically structured transformer encoder to learn the multi-scale features. In addition, the decoder is directly constructed on a lightweight multilayer perceptron, which aggregates information from different layers. In our implementation, the MIT-B5 is taken as its encoder.
- **HRNet+FCN**: In this model, the encoder is the standard HRNet [22]. The decoder is the same as that used in FCN [30]. Please note that only the up-sampling framework of FCN is inherited for segmentation. The standard HRNet is used as its encoder.
- **HRNet+OCR**: It is a SOTA model. The encoder is the standard HRNet [22] and the decoder is the head subnetwork used in OCRNet [34] for segmentation.
- **DyHRNet (Our)**: The pipeline of our method consists of three stages. First, we perform step 3 in Algorithm 1 to train the completely connected network for several epochs to obtain a good initialization. Second, steps 4–11 in Algorithm 1 are implemented to search from the dense connections and obtain channel-wise attention. Third, the final architecture is re-trained with all training data for experimental comparison.

In the experiments, all the above nine models to be compared were performed in the experiment settings suggested in the corresponding paper within the Pytorch framework. In addition, the guidance given by the authors in their works is followed to initialize the hyper-parameters. All models are trained with the SGD strategy.

The base learning rate $\eta$ was initially set as 0.01, the momentum is set to 0.9, and the weight decay is taken as 0.004 during iterations. In addition, the "poly" learning rate policy is employed to adjust the learning rate [65]. At the $t$-th iteration, it is taken as

$$\eta_t = \eta(1 - t/T)^{0.9}, \tag{17}$$

where $T$ is the pre-defined total number of iterations in Algorithm 1. In this way, the sequence of iteration points could be smoother for convergence. It was set to be 40,000 for the Vaihingen dataset. For the larger Potsdam and the LoveDA datasets, it was taken as 80,000. The regularization parameters $\lambda$ in Problem (6) are set to be 0.01 in our implementation.

In the experiments, the size of each mini-batch was taken as 8 for the Vaihingen dataset, and 16 for the Potsdam and LoveDA datasets, when training the model parameters via Algorithm 1. The ResNet-101 [64] was pre-trained on the ImageNet dataset [66]. In our implementation, the original HRNet was also pre-trained on this dataset to obtain a good initialization (https://github.com/HRNet/HRNet-Image-Classification/releases/download/PretrainedWeights/HRNet_W48_C_ssld_pretrained.pth (accessed on 26 April 2023)).

To comprehensively evaluate the different models, two overall benchmark metrics were employed, namely the Overall Accuracy (OA) score of the classification and the mean Intersection over Union ($mIoU$) evaluated on the testing samples. In addition, the accuracy for each class will also be reported for comparison.

### 4.3. Experiment Results

To give a comprehensive comparison between the models, we list the quantitative scores of $OA$ and $mIoU$ obtained by the ten models on the Vaihingen, Potsdam, and LoveDA datasets in the right panels in Tables 1–3, respectively. All these values are achieved on the corresponding test images and averaged as a whole on the categories. Specifically, two methods are implemented to output the final scores. One is to calculate them directly on the images in the testing dataset. Another is to evaluate them via the flip and multi-scale (MS) testing, which means that the testing images are randomly flipped and/or resized to augment the samples.

As can be seen from the comparative results in these tables, our model DyHRNet largely outperforms the seminal FCN and the famous UNet, which were initially developed for semantic segmentation. It is also superior to the powerful PSPNet, DeepLabV3+, and OCRNet models, which were all designed with the tricks of multi-scale information fusion in large receptive fields. In addition, it outperforms the SOTA models, including SETR and SegFormer, in which the frontier technique of Transforms is employed to construct the models. In parallel, our model is developed from the original HRNet. The two combinations, HRNet+FCN and HRNet+OCR, were compared against our model. By contrast, the HRNet+FCN employs a simple head network for semantic segmentation, while the HRNet+OCR uses a complex head network for this task. As can be seen from these tables, our model achieves better results, demonstrating its effectiveness for RS image segmentation.

**Table 1.** Quantitative comparison results on the Vaihingen testing set. The digits are the percent scores (%). [†] means that the scores are obtained via the flip and MS testing. (Bold font represents the highest performance of the class)

| Method | Imp. Surf. | Building | Low Veg. | Tree | Car | OA | mIoU | OA [†] | mIoU [†] |
|---|---|---|---|---|---|---|---|---|---|
| FCN [30] | 84.99 | 91.31 | 70.31 | 79.57 | 76.18 | 89.76 | 80.47 | 90.34 | 81.87 |
| UNet [31] | 82.78 | 87.41 | 67.69 | 78.17 | 65.90 | 88.15 | 76.39 | 89.44 | 79.15 |
| PSPNet [32] | 85.83 | 91.49 | 71.37 | 79.90 | 75.14 | 90.16 | 80.75 | 90.76 | 82.38 |
| DeepLabV3+ [33] | 86.20 | 91.61 | 71.43 | 79.74 | 75.18 | 90.23 | 80.83 | 90.81 | 82.19 |
| OCRNet [34] | 84.70 | 90.57 | 69.74 | 78.83 | 66.71 | 89.36 | 78.11 | 90.23 | 79.99 |
| SETR [35] | 83.12 | 88.21 | 67.07 | 77.86 | 55.31 | 88.13 | 74.31 | 89.04 | 75.72 |
| SegFormer [36] | 86.72 | **92.42** | 72.30 | 80.53 | 78.54 | 90.68 | 82.10 | 91.07 | 83.06 |
| HRNet+FCN [30] | 85.91 | 91.91 | 71.03 | 79.90 | 76.40 | 90.17 | 81.03 | 90.75 | 82.35 |
| HRNet+OCR [34] | 86.77 | 91.43 | 73.51 | 80.65 | 78.34 | 90.54 | 82.14 | 91.48 | 83.73 |
| **DyHRNet (Ours)** | **87.06** | 92.26 | **73.68** | **80.83** | **82.76** | **90.96** | **83.32** | **91.70** | **84.34** |

**Table 2.** Quantitative comparison results on the Potsdam testing set. The digits are the percent scores (%). † means that the scores are obtained via the flip and MS testing. (Bold font represents the highest performance of the class)

| Method | Imp. Surf. | Building | Low Veg. | Tree | Car | OA | mIoU | OA † | mIoU † |
|---|---|---|---|---|---|---|---|---|---|
| FCN [30] | 87.00 | 93.58 | 75.77 | 78.90 | 92.44 | 90.44 | 85.54 | 90.82 | 86.23 |
| UNet [31] | 83.63 | 89.08 | 73.28 | 77.75 | 89.69 | 88.36 | 82.69 | 89.14 | 84.02 |
| PSPNet [32] | 87.44 | 94.03 | 76.64 | 79.33 | 93.02 | 90.80 | 86.09 | 91.29 | 86.81 |
| DeepLabV3+ [33] | 87.40 | 93.82 | 76.60 | 79.28 | 93.03 | 90.80 | 86.03 | 91.27 | 86.72 |
| OCRNet [34] | 85.17 | 90.22 | 75.31 | 76.96 | 89.83 | 89.33 | 83.50 | 90.21 | 84.92 |
| SETR [35] | 78.28 | 84.78 | 66.60 | 66.17 | 77.33 | 84.25 | 74.63 | 85.54 | 76.77 |
| SegFormer [36] | 87.54 | 94.04 | **78.15** | 80.06 | 92.22 | 91.18 | 86.40 | 91.61 | 87.19 |
| HRNet+FCN [30] | 81.22 | 93.75 | 76.86 | 79.54 | 92.65 | 90.80 | 84.80 | 91.36 | 86.89 |
| HRNet+OCR [34] | 86.31 | 93.03 | 77.04 | 78.86 | 90.58 | 90.57 | 85.16 | 91.25 | 86.62 |
| **DyHRNet (Ours)** | **87.77** | **94.07** | 77.93 | **80.59** | **93.05** | **91.20** | **86.68** | **91.79** | **87.56** |

**Table 3.** Quantitative comparison results on the LoveDA testing set. The digits are the percent scores (%) . † means that the scores are obtained via the flip and MS testing. Here, "Back." stands for "Background", "Build." stands for "Building", "Agri." stands for "Agriculture". (Bold font represents the highest performance of the class)

| Method | Back. | Build. | Road | Water | Barren | Forest | Agri. | OA | mIoU | OA † | mIoU † |
|---|---|---|---|---|---|---|---|---|---|---|---|
| FCN [30] | 52.75 | 62.63 | 53.62 | 66.06 | 22.38 | 38.97 | 49.54 | 68.11 | 49.42 | 67.47 | 42.66 |
| UNet [31] | 49.06 | 57.82 | 47.87 | 47.69 | 25.90 | 37.66 | 43.65 | 63.23 | 44.24 | 64.05 | 44.04 |
| PSPNet [32] | 55.14 | 64.24 | 55.54 | 68.03 | 27.01 | 41.56 | 51.53 | 70.27 | 51.86 | 69.98 | 51.34 |
| DeepLabV3+ [33] | 54.19 | 64.39 | 55.67 | 68.14 | 27.17 | 41.44 | 49.29 | 69.50 | 51.47 | 69.60 | 51.32 |
| OCRNet [34] | 53.10 | 51.79 | 54.56 | 59.71 | 23.70 | 35.69 | 46.99 | 66.56 | 46.51 | 65.54 | 45.21 |
| SETR [35] | 47.98 | 57.24 | 40.37 | 59.70 | 20.23 | 39.54 | 36.39 | 62.50 | 43.06 | 62.01 | 42.65 |
| SegFormer [36] | 52.82 | **65.50** | **56.63** | 70.64 | 29.29 | **41.63** | 51.93 | 69.96 | 52.63 | 70.07 | 52.25 |
| HRNet+FCN [30] | 54.37 | 60.97 | 56.08 | 68.54 | 26.77 | 41.11 | 52.09 | 69.87 | 51.42 | 70.06 | 51.64 |
| HRNet+OCR [34] | **54.62** | 63.47 | 52.76 | 70.54 | 34.68 | 35.55 | 51.02 | 69.89 | 51.81 | 69.24 | 50.27 |
| **DyHRNet (Ours)** | 54.03 | 62.83 | 55.82 | **72.31** | **35.67** | 38.04 | **58.92** | **71.55** | **53.95** | **70.98** | **53.72** |

Furthermore, most images render class imbalance at the pixel level in these datasets, i.e., some objects (for example, buildings) occupy large regions, while the small objects (for example, cars) have small regions here and there in images. Thus, we employ the metric *mIoU* to measure the goodness of the segmentation, respectively, on the category level. As can be seen from the left panels in Tables 1–3, our model achieves better scores for most categories in these datasets. It renders a significant performance enhancement on small objects. Table 1 shows that DyHRNet achieves 4.42% higher accuracy than the second model (SegFormer) on the car category in the Vaihingen dataset. Such an enhancement can also be witnessed in Table 2 on the tiny objects, including the car and the tree in the scenes. In addition, as witnessed in Table 3, our model also obtains the SOTA performance on the LoveDA dataset.

Figure 5 illustrates the radar charts on the three datasets to further compare the performances of the ten models, category by category. The points in these charts stand for the corresponding *mIoU* scores, which are obtained via data augmentation (flip and MS testing) tricks on the testing dataset. From these figures, it is seen that the curves obtained by our model always locate at the outer region, indicating that it achieves higher performance compared with the nine models.

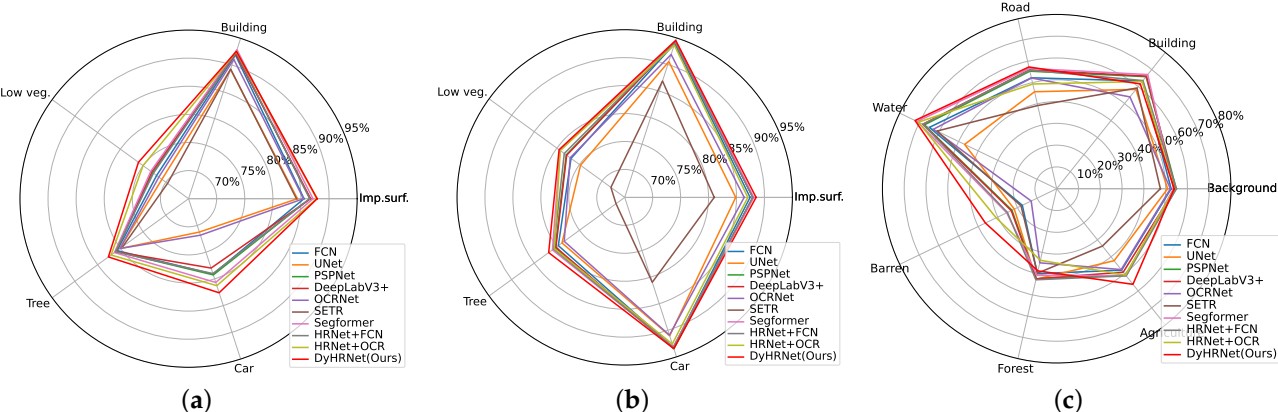

**Figure 5.** Comparisons category by category on the three datasets via Radar chart. The digits are the *mIoU* scores, obtained via the flip and MS testing. (**a**) Vaihingen; (**b**) Potsdam; (**c**) LoveDA.

Finally, Figures 6–8 demonstrate the segmentation results of a few images obtained by the ten models, including FCN, UNet, PSPNet, DeepLabV3+, OCRNet, SETR, SegFormer, HRNet+FCN, HRNet+OCR and DyHRNet.

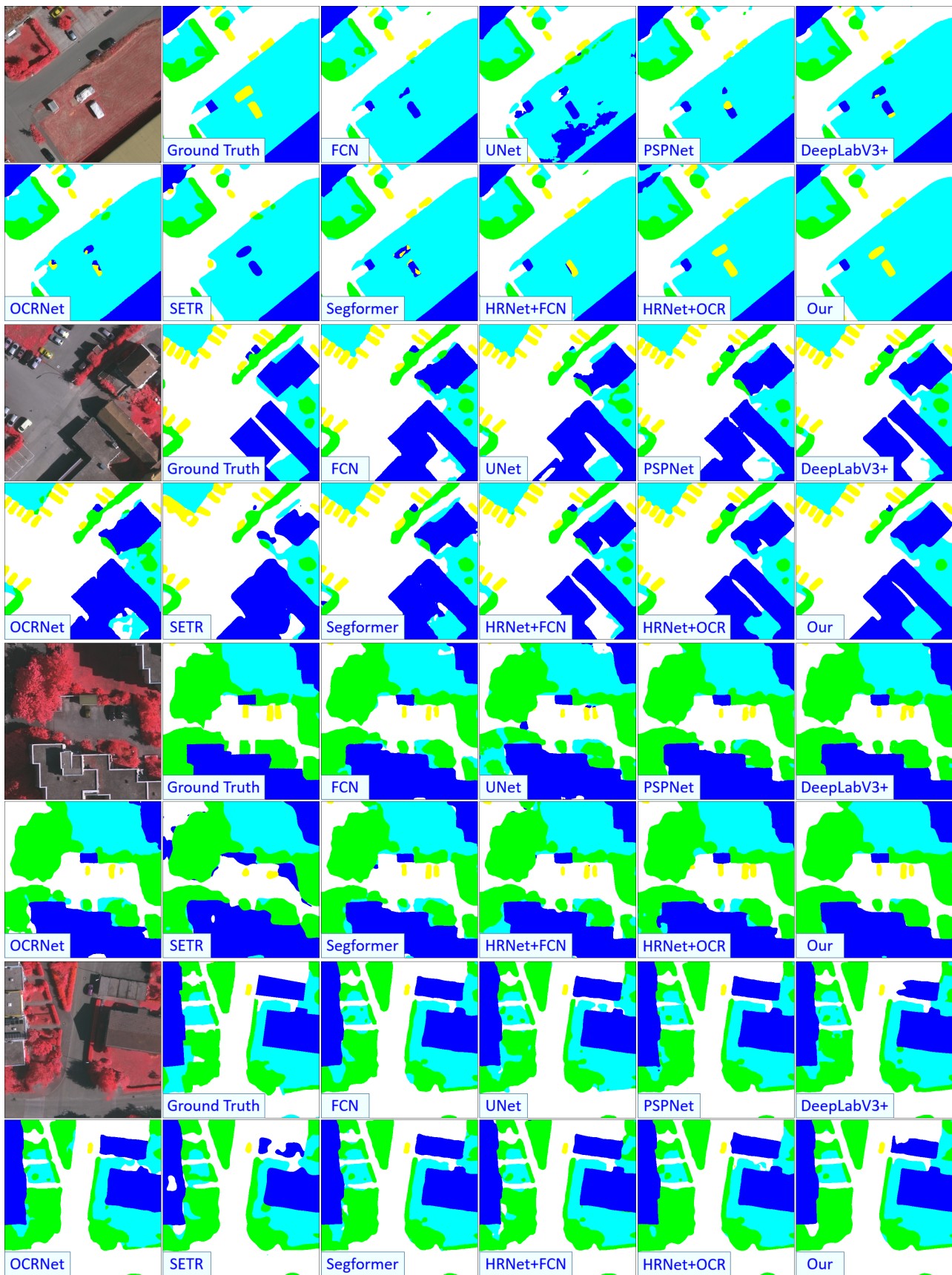

**Figure 6.** Visual comparisons between our method and other related methods on the Vaihingen dataset. The label includes six categories: impervious surface (white), building (blue), low vegetation (cyan), tree (green), and car (yellow).

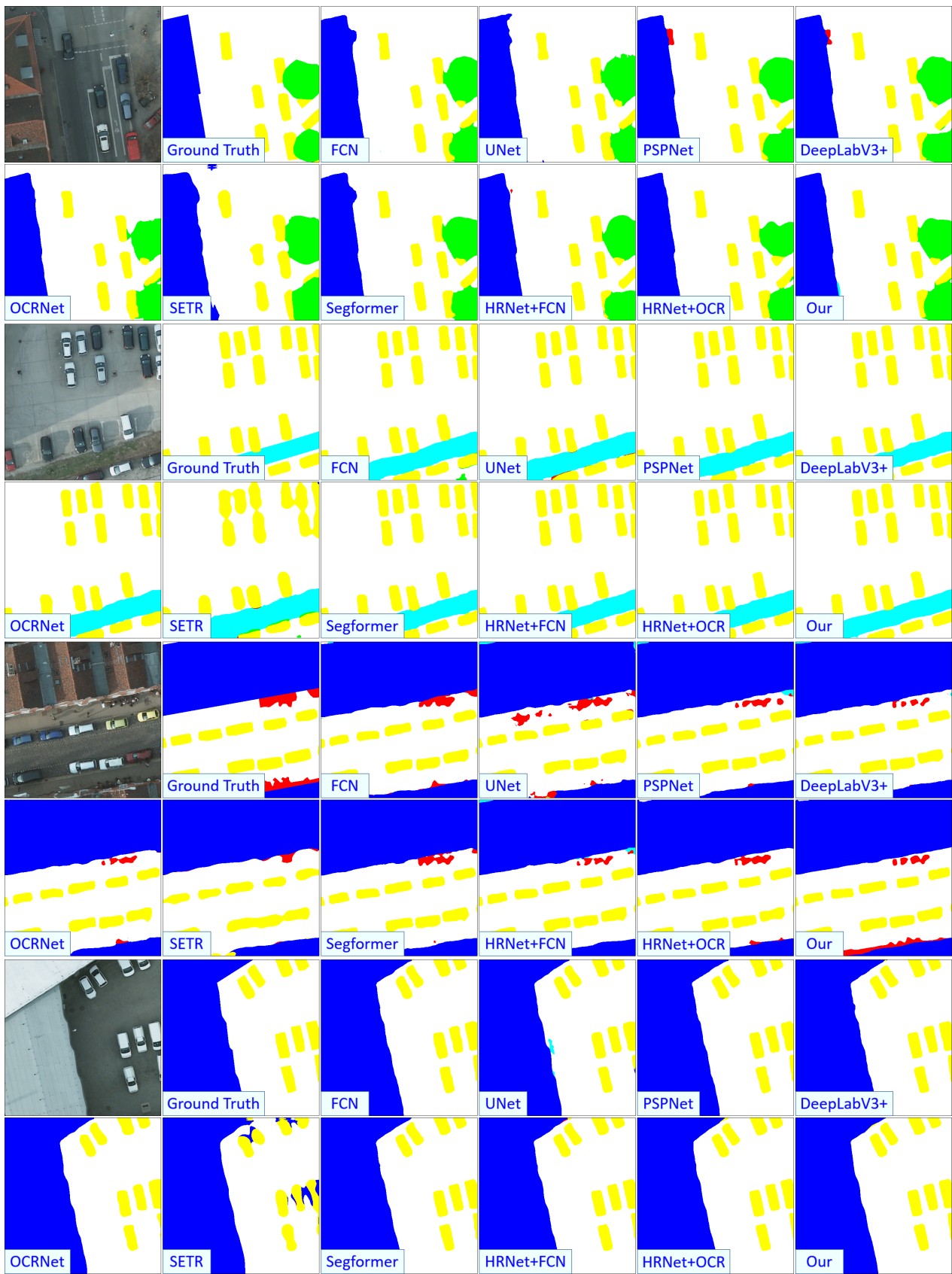

**Figure 7.** Visual comparisons between our method and other related methods on the Potsdam dataset. The label includes six categories: impervious surface (white), building (blue), low vegetation (cyan), tree (green), car (yellow), and clutter/background (red). Here, backgrounds are directly shown as they were considered to be masks when training the models.

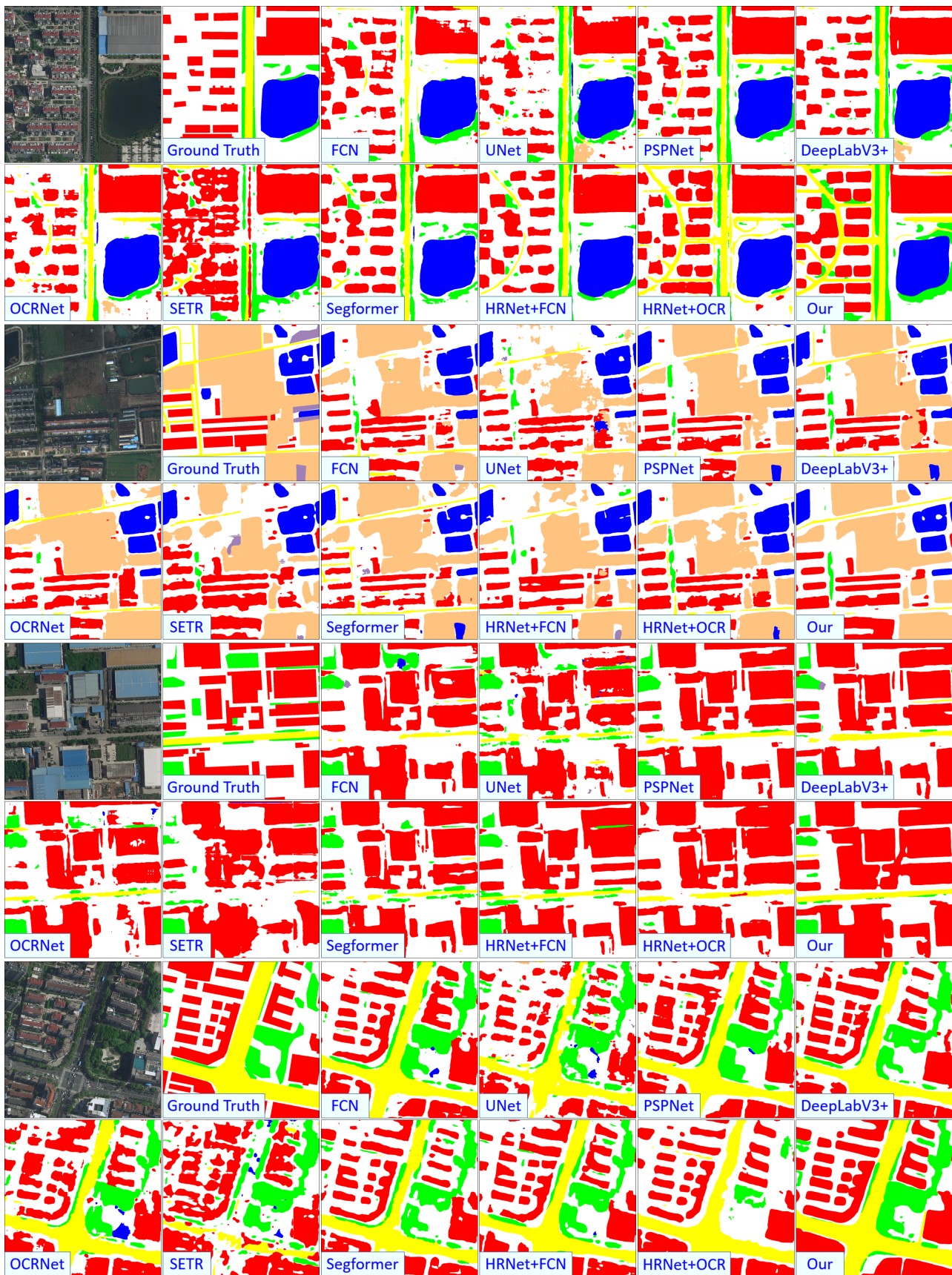

**Figure 8.** Visual comparisons between our method and other related methods on the LoveDA dataset. The label includes six categories: Background (white), Building (red), Road (yellow), Water (blue), Barren (plum), Forest (green), and Agriculture (orange).

As can be seen from these figures, the FCN and UNet render limited performances in segmenting the tiny objects. For the segmentation results obtained by the PSPNet, the boundaries of objects do not keep well, which can be seen from those regions of some buildings in the LoveDA dataset. DeepLabV3+, OCRNet, SETR, SegFormer, HRNet+FCN, and HRNet+OCR indeed improve the quality of segmentation, but false negatives can also be perceived from the segmentation results. This fact can be witnessed clearly in the Vaihingen dataset. In contrast, our model performs better and shows satisfactory edge preserving, typically on the Vaihingen and Potsdam datasets, where coherent segmentations are obtained on fine-structured buildings and tiny objects. In addition, on the challenging LoveDA dataset, for example, in the first image in Figure 8, the roads in the left region are all manually labeled as background, but our model segments well for these regions.

In summary, the above comparisons show that our DyHRNet is capable of segmenting confusing artificial objects in high-resolution RS images. In addition, it also shows good quality segmentations with satisfactory edge preserving for confusing size-variable objects and tiny objects such as cars and trees without performing post-processing. This indicates that our DyHRNet learned from the densely connected HRNet with channel-wise attention has powerful data adaptability for RS images.

### 4.4. Ablation Study

This subsection reports the ablation experiments to evaluate the importance of different components proposed in our method. Please note that in our work, there are two fundamental designs. One is the dynamic dense connections achieved with the APG algorithm in Section 3.2, and another is the dynamic channels in Section 3.3. Table 4 reports four combinations with or without using these two designs on the Vaihingen dataset. It is seen that performing both fundamental modelings helps enhance the performance. This indicates the validation of our proposed approach.

When learning our DyHRNet, there are two subtasks: training the neural network and searching from the dense connections. They are solved alternatively by fixing one for another in Algorithm 1. To further evaluate the importance of the APG algorithm for selecting out the important connections, we conducted another group of ablation experiments, in which the APG algorithm is performed once a time every two or four times during iterations. Table 5 reports the performances obtained in this experimental setting. It is seen that the performances decrease in most cases when the interval number increases, compared with the original step-by-step for these two subtasks. This indicates that the APG algorithm for sparse selection from the dense connections plays a significant role in data-driven learning for the semantic segmentation of RS images.

**Table 4.** The ablation study of our proposed DyHRNet on the Vaihingen dataset with different combinations. The digits are the percent scores (%). † means that the scores are obtained via the flip and MS testing.

| Channel-Wise Attention | APG Sparse Optimization | OA | mIoU | OA † | mIoU † |
|:---:|:---:|:---:|:---:|:---:|:---:|
| ✗ | ✗ | 90.54 | 82.14 | 91.48 | 83.73 |
| ✓ | ✗ | 90.92 | 83.11 | 91.55 | 84.12 |
| ✗ | ✓ | 90.94 | 83.23 | 91.54 | 84.15 |
| ✓ | ✓ | 90.96 | 83.32 | 91.70 | 84.34 |

**Table 5.** The ablation study of our proposed DyHRNet on the Vaihingen dataset. The digits are the percent scores (%). † means that the scores are obtained via the flip and MS testing.

| Train/Search Step | OA | mIoU | OA † | mIoU † |
|:---:|:---:|:---:|:---:|:---:|
| 1/1 | 90.96 | 83.32 | 91.70 | 84.34 |
| 2/1 | 90.46 (↓) | 81.29 (↓) | 91.20 (↓) | 82.66 (↓) |
| 4/1 | 90.40 (↓) | 81.46 (↓) | 90.77 (↓) | 81.58 (↓) |

Finally, the DyHRNet is used as the backbone of the models with different heads for semantic segmentation. Specifically, the heads in FCN and OCR are combined with the DyHRNet, respectively, for experimental evaluation. This treatment generates two models, known as "DyHRNet+FCN" and "DyHRNet+OCR". The latter is the model used in Section 4.2. For convenience, the original HRNet is also employed for comparison. The experimental settings are the same used in Section 4.2. Table 6 reports the experimental results on the Vaihingen dataset. It is seen that the performance of "DyHRNet+FCN" is better than that of "HRNet+FCN", while the performance of "DyHRNet+OCR" is better than that of "HRNet+OCR". As our model is learned from the original HRNet, this fact indicates the usage of our proposed method.

**Table 6.** The ablation study of the original HRNet and our DyHRNet with different segmentation heads on the Vaihingen dataset. The digits are the percent scores (%). [†] means that the scores are obtained via the flip and MS testing.

| Backbone | FCN | OCR | OA | mIoU | OA [†] | mIoU [†] |
|---|---|---|---|---|---|---|
| HRNet | ✓ | | 90.17 | 81.03 | 90.75 | 82.35 |
| | | ✓ | 90.54 | 82.14 | 91.48 | 83.73 |
| DyHRNet | ✓ | | 90.80 | 82.72 | 91.45 | 83.82 |
| | | ✓ | 90.96 | 83.32 | 91.70 | 84.34 |

### 4.5. Computational Efficiency

Here we analyze the computational efficiency of the models. To give a comprehensive analysis, the following models are compared, including the FCN, UNet, PSPNet, DeepLabV3+, OCRNet, SETR, SegFormer, HRNet+FCN, HRNet+OCR, DyHRNet +FCN, and DyHRNet+OCR. In Table 7, the performance of the architecture with a combination of the learned backbone DyHRNet and the head of the FCN, namely "DyNRnet+FCN", is additionally reported for comparison. The factors related to the computational efficiency are listed in Table 7, including the number of the parameters and the number of the FLoating-point OPerations (FLOPs) with Giga Multiplier ACcumulators (GMACs) in the model.

**Table 7.** Computational efficiency, including the total number of the FLoating-point OPerations (FLOPs) and the total number of the parameters in the model. The backbones and the mIoU scores calculated on the Vaihingen testing dataset are also listed here for comparison.

| Method | Backbone | #Params | GFLOPs | mIoU (%) |
|---|---|---|---|---|
| FCN [30] | ResNet-101 | 68.48M | 275.38 | 80.47 |
| PSPNet [32] | ResNet-101 | 67.96M | 256.14 | 80.75 |
| DeepLabV3+ [33] | ResNet-101 | 62.57M | 253.93 | 80.83 |
| OCRNet [34] | ResNet-101 | 55.51M | 230.57 | 78.11 |
| UNet [31] | UNet-S5-D16 | 29.06M | 202.63 | 76.39 |
| SETR [35] | VIT-L | 318.45M | 260.85 | 74.31 |
| SegFormer [36] | MIT-B5 | 82.01M | 52.45 | 82.10 |
| HRNet+FCN [30] | HRNetV2-W48 | 65.85M | 93.43 | 81.03 |
| HRNet+OCR [34] | HRNetV2-W48 | 70.36M | 162.21 | 82.14 |
| DyHRNet+FCN | DyHRNet | 63.88M | 91.57 | 82.72 |
| DyHRNet+OCR | DyHRNet | 66.44M | 158.81 | 83.32 |

In addition, the backbones and the mIoU scores calculated on the Vaihingen testing dataset are also listed for comparison. In the experiments, the FCN, PSPNet, DeepLabV3+, and OCRNet all take the "ResNet-101" as their backbone. By contrast, UNet, SETR, SegFormer, and HRNet all have their own backbone. It is worth pointing out that the downsampling operations in UNet and ResNet are different from each other. Furthermore, they

also have different output channels in consecutive stages. Thus, the "ResNet-101" is not selected as the backbone for UNet in our experiments.

In general, the number of parameters and the number of the FLOPs are two important factors to evaluate the computation scale in deep models. In particular, FLOPs can directly reflect the computational complexity of the model, which is related to the size of the input image and the neural architecture. As can be seen in Table 7, the computation scale in the DyHRNet was reduced to a medium level, but it achieves the best performance on the Vaihingen dataset. For example, by contrast to the FCN, our model has the parameters at the same level, but the computational scale is largely reduced to 57.6%. In addition, from Table 7, it is seen that the numbers of parameters and FLOPs in DyHRNet+FCN are both smaller than those in HRNet+FCN. This fact can also be witnessed when HRNet+OCR and DyHRNet+OCR are compared to each other. Thus, it can be concluded that the decrease of the computations in our model occurs in the backbone. This is due to the architecture learning from the HRNet, where those cross-resolution connections and channels with zero contributions will be ignored. This indicates the effectiveness of our method.

## 5. Discussions

### 5.1. The Learned Structure and Iteration Process Analysis

Algorithmically, the main task in this study is to solve the optimization problem in (6). In this task, the key job is to search for the important connections among the eight groups of dense connections in the original HRNet. This is achieved using the APG algorithm with sparse selection tricks. Figure 9a–c illustrate the learned weights of the eight groups of dense connections. For example, in the first panel in Figure 9a–c, there is a $2 \times 2$ matrix, including four weights. This panel corresponds to the group of dense connections in stage 2 in Figure 1. For clarity, we take Figure 9a as an example to explain the details. In the first panel, the value "0.021" is the learned weight of the connection between $R_{1,1}$ and $O_{1,2}$, "0.026" is that of the connection between $R_{1,1}$ and $O_{2,2}$, "0.000" is that of the connection between $R_{2,1}$ and $O_{1,2}$, and "0.202" is that of the connection between $R_{2,1}$ and $O_{2,2}$. Based on Figure 9a–c, it is seen that there are many connections with weights tending to zero. This fact indicates that our search approach plays an active role in the original architecture of HRNet, helping enhance the performance of the model for semantic segmentation with end-to-end training.

Based on the visualizations in Figure 9a–c, one interesting phenomenon is that all the connections at the same horizontal level retain relatively higher weights. This means that the feature maps at the same resolution should be maintained, which is more important for nonlinear feature learning. Small weights are always learned from cross-resolutions, and most of them correspond to the connections from high to low resolution. This may be because there are many tiny objects in the RS images, avoiding discarding the details of the tiny objects and performing those down-sampling operations.

As demonstrated in Figure 1, a total of 88 connections are employed as candidates to be selected by the APG algorithm described in Section 3.2. Figure 10 shows the *mIoU* scores and the total weights of the 88 connections on the Vaihingen, Potsdam, and LoveDA datasets in the iterations. It is seen that the *mIoU* scores of the learned models after 10,000 iterations will stay at the same level without rendering significant changes. However, the total sum of the learned weights is drastically decreased along with the increase of iterations. In the beginning, all the weights are taken as 1.0. Thus, there is a point marked as "88.0" in each figure. Then, it reduces and converges to keep at a stationary level. This means that the original dense connections have different contributions, far below the identical ones with the same importance. This fact also indicates that the connections with zero weights can be deleted from the original HRNet in the way of end-to-end learning.

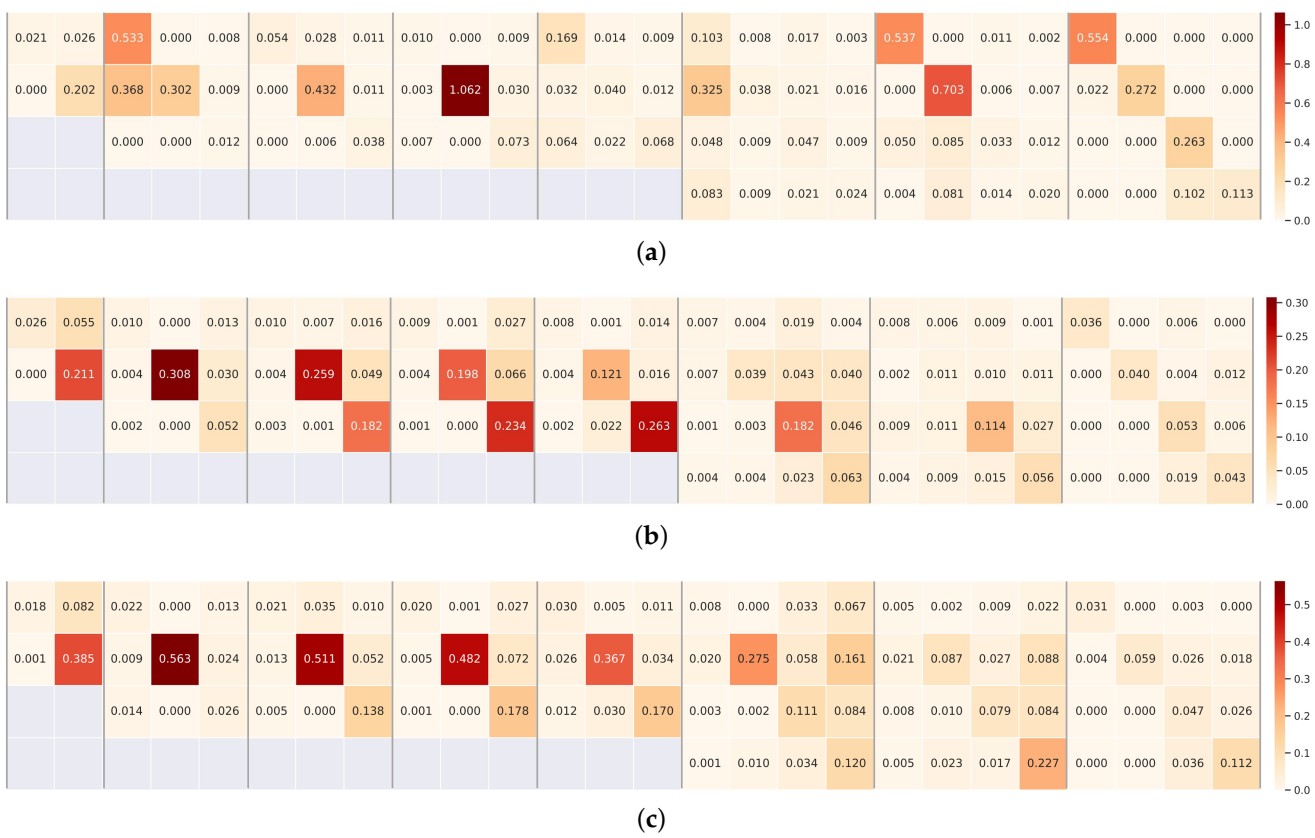

**Figure 9.** Visualization of the obtained weights of the dense connections. In each panel separated by vertical lines, there are a group of weights, corresponding to the connections in Figure 1. The larger the weight, the more important the connection; (**a**) Vaihingen; (**b**) Potsdam; (**c**) LoveDA.

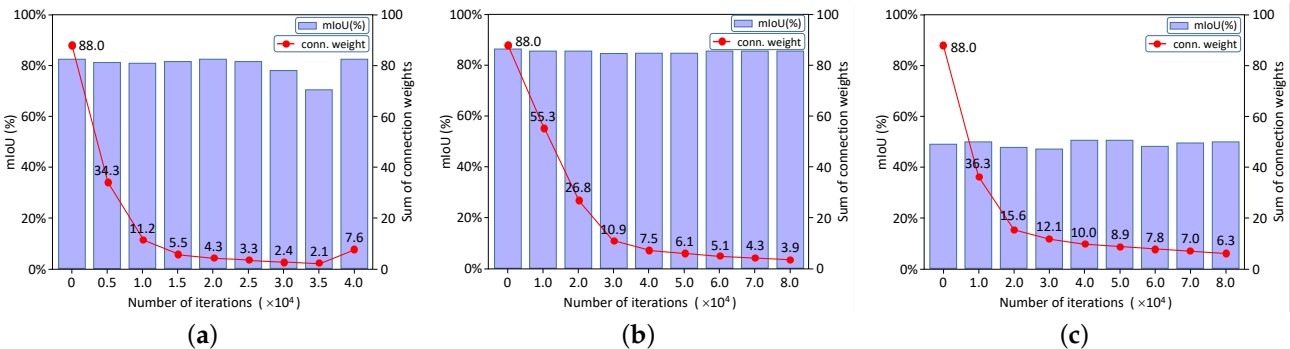

**Figure 10.** The mIoU score and the overall weight sum learned from the dense connections with iterations; (**a**) Vaihingen; (**b**) Potsdam; (**c**) LoveDA.

### 5.2. The Behavior of the Accelerated Proximal Gradient Algorithm

In Algorithm 1, one of the main tasks is to identify the important cross-resolution connections in the original HRNet for semantic segmentation. To this end, the APG algorithm is employed to solve the sparse regularization subproblem in Problem (6). Algorithmically, it starts with the original HRNet to modulate the weights of the cross-resolution connections, which are all initialized as 1.0 for iterations. This gives an equal chance for all the connections to be evaluated. In this way, a top-down training strategy is performed, in which all the weights are gradually nullified to small scores. After training, the connections with zero weights will be discarded for prediction.

Alternatively, a bottom-up training strategy could be implemented for the APG algorithm. To this end, we first assign a small weight to the connections, and then pre-train

the original HRNet on the ImageNet dataset. Then, Algorithm 1 is implemented. Table 8 reports the experimental results on the Vaihingen dataset. Specifically, in Table 8, "Init-0.1", "Init-0.5", and "Init-1.0" correspond to the cases with all weights initialized to 0.1, 0.5, and 1.0, respectively. In this group of experiments, the maximum number of iterations is taken as 40,000, and all the training samples described in Subsection 4.1 are taken to learn the model. In the experiments, it is observed that small initial weights indeed help speed up the convergence, but the performance decreases drastically. As can be seen from Table 8, the models trained with small initial weights perform unsatisfactorily, compared to that with all weights set to be 1.0 for learning.

**Table 8.** Quantitative comparison results on the Vaihingen testing set with different initial weights to the cross-resolution connections for the AGP algorithm. The digits are the percent scores (%). $^\dagger$ means that the scores are obtained via the flip and MS testing.

| Weight Initialization | Imp. Surf. | Building | Low Veg. | Tree | Car | OA | mIoU | OA $^\dagger$ | mIoU $^\dagger$ |
|---|---|---|---|---|---|---|---|---|---|
| Init-0.1 | 75.85 | 84.48 | 65.93 | 74.19 | 55.74 | 85.63 | 71.24 | 82.46 | 66.60 |
| Init-0.5 | 83.06 | 87.41 | 68.75 | 78.26 | 61.39 | 88.38 | 75.77 | 89.03 | 77.01 |
| Init-1.0 | 87.06 | 92.26 | 73.68 | 80.83 | 82.76 | 90.96 | 83.32 | 91.70 | 84.34 |

To further investigate the behavior of the AGP algorithm, experiments with different ratios of training data and different numbers of iterations are conducted on the Vaihingen dataset. The goal is to demonstrate whether the weights of the cross-resolution connections learned by the AGP algorithm change drastically. Specifically, in the first group of experiments, the ratios are set as 10%, 50%, and 100% of all the total training samples, respectively. In this group, the maximum number of iterations, namely parameter $T$ in Algorithm 1, is taken as 40,000. In another group of experiments, $T$ is set as 10,000, 20,000, and 40,000, respectively. In this case, all the training samples are employed to learn the model. In this group, the weights of the cross-resolution connections are initialized to 1.0 for the APG algorithm. Figure 11 visualizes the weights of the cross-resolution connections in the DyHRNet, which are obtained, respectively, with these experimental settings. It is seen that all the weights are dropped below 1.0. Table 9 reports the performances of the learned models. By considering together the weights visualized in Figure 9a and the performance scores, which are obtained with 100% training samples and $T = 40,000$, there are no significant changes both in performance scores and in weight values.

**Table 9.** Quantitative comparisons on the Vaihingen testing set with different ratios of training samples and different numbers of iterations. The digits are the percent scores (%). $^\dagger$ means that the scores are obtained via the flip and MS testing. Here, "100% + 40,000" means the model is learned with 100% of the training samples, and the maximum number of iterations ($T$) in Algorithm 1 is set to be 40,000.

| | Imp. Surf. | Building | Low Veg. | Tree | Car | OA | mIoU | OA $^\dagger$ | mIoU $^\dagger$ |
|---|---|---|---|---|---|---|---|---|---|
| #training (10%) | 86.49 | 91.81 | 71.51 | 79.72 | 79.11 | 90.41 | 81.73 | 91.10 | 82.07 |
| #training (50%) | 86.80 | 91.85 | 72.74 | 80.39 | 80.39 | 90.70 | 82.43 | 91.41 | 83.57 |
| $T = 10,000$ | 86.31 | 92.00 | 72.46 | 80.68 | 81.75 | 90.68 | 82.64 | 91.37 | 83.84 |
| $T = 20,000$ | 87.53 | 92.70 | 73.28 | 80.83 | 82.16 | 91.10 | 83.30 | 91.55 | 84.11 |
| 100% + 40,000 | 87.06 | 92.26 | 73.68 | 80.83 | 82.76 | 90.96 | 83.32 | 91.70 | 84.34 |

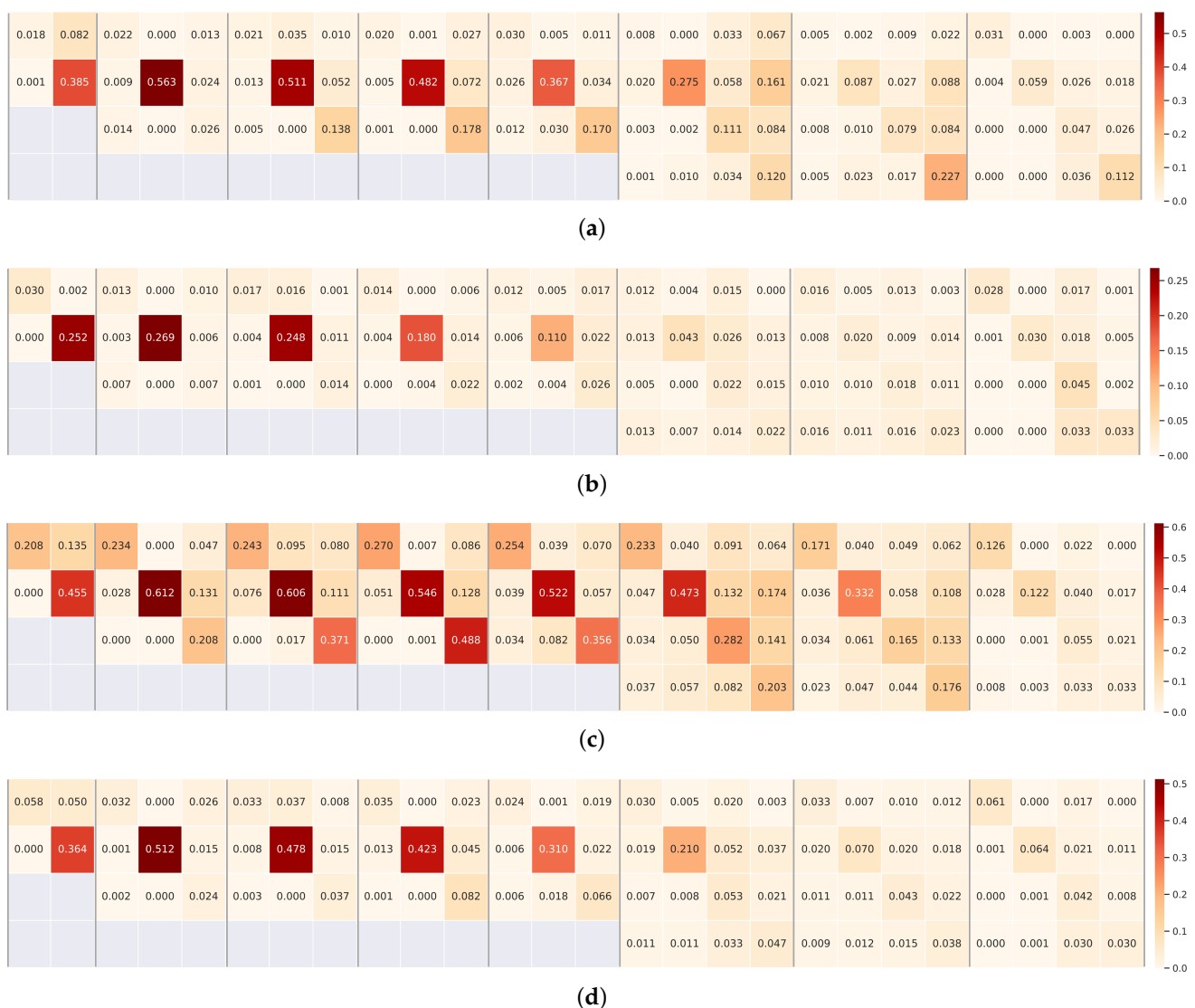

**Figure 11.** Visualization of the weights of the dense connections learned from the Vaihingen dataset with different settings. In each panel separated by vertical lines, there are a group of weights, corresponding to the connections in Figure 1. The weights in (**c**,**d**) are learned with all of the training samples used in Table 1. (**a**) The weights learned with 10% training samples; (**b**) The weights learned with 50% training samples; (**c**) The weights learned with 10,000 iterations; (**d**). The weights were learned with 20,000 iterations.

### 5.3. Implications and Limitations

In this study, we have conducted experimental evaluations on the three challenging public datasets. In these RS images, many artificial objects with different sizes and confusing appearances are located here and there. Achieving consistent and accurate semantic segmentation is a challenging task. The primary function of the DyHRNet is to enhance the quality of semantic segmentation via cross-resolution feature fusion. To this end, the structure of the original HRNet has been exploited by evaluating the contributions of the dense connections and the channels related to the cross-resolution feature fusion. With the problem formulation addressed under the NAS framework with channel-wise attention, the goal is well achieved. This means that the structure in the HRNet with parallel streams of high-, medium-, and low-resolution representation attends well to the needs of segmenting RS objects of multi-scales. In addition, compared with the original HRNet, the reduced parameters in our DyHRNet indicate that not all the connections and channels contribute equally to the task. In other words, redundant connections and channels exist

for performing the cross-resolution feature fusion. Therefore, one can consider designing lightweight or dynamic HRNet or more general architectures by keeping the advantages of the multi-scale structure design rendered in the HRNet for this or similar tasks in the fields of RS image processing.

Methodologically, the proposed DyHRNet renders enhancements on both architecture design and segmentation performance. However, it has several limitations, which are described as follows:

- As outlined in Table 7, our model has more than 66 million parameters, and the computational scale is up to about 158 GFLOPs. Thus, high-performance computing resources with GPUs are needed to fulfill the computing task. This indicates that releasing it on edge computing devices is difficult with its current version. However, the sum of the total weights demonstrated in Figure 10 indicates many connections with small contributions. Thus, the scale of the models could be reduced by model pruning.
- The performance of the DyHRNet could be further improved for the objects with rich visual appearances and those with blurring edges, for example, the buildings and forests in the LoveDA dataset. On the one hand, more training samples are needed to guarantee the generalization of the model. On the other hand, some prior knowledge with constrained forms or regularization terms in the loss function could be introduced to guide the model training.
- The current version of the DyHRNet is not general given the dynamic architecture design. This is because the tricks with NAS are only applied to dense connections. However, many convolutional operations are contained in the blocks $\mathbf{O}_{i,j}$ in Figure 1. This indicates that one can perform the NAS on all operations to learn more general architecture for different needs in practice.

## 6. Conclusions

This work proposes a Dynamic High-Resolution Network (DyHRNet) for the semantic segmentation of RS images. The DyHRNet is an architecture-learnable model under a neural architecture search (NAS) framework with channel-wise attention. It takes the primary HRNet as its super-architecture, which has four parallel streams to retain the different resolutions for multi-scale feature fusion simultaneously. The learning task has been explicitly formulated with a series of sparse regularizations, where the Accelerated Proximal Gradient (APG) algorithm is introduced to solve the sparse optimization model. In contrast to the static HRNet, the dynamic merits of the DyHRNet with data adaptability lie in the following two aspects. On the one hand, the structure of the DyHRNet is dynamically adjusted for cross-resolution feature fusion by identifying those unimportant connections and ignoring those with zero contributions in the original HRNet. On the other hand, the contributions of channel-wise contribution for feature fusion are modulated automatically to enhance the representation capability of the proposed model.

Extensive experiments have been conducted on three public challenging RS image datasets. Nine classical or SOTA models have been employed to compare with our model. Comparative experiment results with numerical scores, visual segmentations, the learned structures, the iteration process analysis, and the ablation study demonstrate the advantages and effectiveness of the proposed DyHRNet. In the future, we aim to design a lightweight HRNet to perform the semantic segmentation of RS images in edge computing devices.

**Author Contributions:** Conceptualization, S.G. and S.X.; methodology, Q.Y. and S.X.; software, Q.Y. and S.G.; validation, S.G., Q.Y., S.X., P.W. and X.W.; investigation, S.G. and S.X.; writing—original draft preparation, S.G. and S.X.; writing—review and editing, S.G., S.X. and P.W.; formal analysis, S.G.; visualization, S.G. and Q.Y.; data curation, Q.Y.; supervision, S.X., P.W. and X.W.; resources, P.W. and X.W.; project administration, P.W., X.W.; funding acquisition, X.W. All authors have read and agreed to the published version of the manuscript.

**Funding:** This work was supported by Key Research Program of Frontier Sciences, CAS (grant number: ZDBS-LY-DQC016), National Key Research and Development Program of China(grant number: 2022YFF1301803), and National Natural Science Foundation of China (NSFC) under grant 62076242.

**Data Availability Statement:** Three public datasets (i.e., the Vaihingen, Potsdam, LoveDA) were included in this study. Both the Vaihingen dataset and the Potsdam dataset were obtained via the official website: https://www.isprs.org/education/benchmarks/UrbanSemLab/default.aspx (accessed on 26 April 2023). The LoveDA dataset was downloaded from the webpage: https://github.com/Junjue-Wang/LoveDA (accessed on 26 April 2023).

**Conflicts of Interest:** The authors declare no conflict of interest.

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
