# Peer review of "Dynamic High-Resolution Network for Semantic Segmentation in Remote-Sensing Images"

_remotesensing, doi:10.3390/rs15092293_

Round 1

Reviewer 1 Report

The authors proposed a Dynamic High-Resolution Network (DyHRNet) for the semantic segmentation of remote sensing images in the manuscript. The proposed DyHRNet is an architecture-learnable model under Neural Architecture Search (NAS) framework with channel-wise attention. The authors conducted extensive experiments for evaluation. The manuscript is well-written, and the proposed methods sound convincing. However, there are some suggestions.

1. The idea of using NAS for dynamic heads has been widely investigated in the current works. It would be better if the authors could discuss and mention them in the related work section.

a. Nas-unet: Neural architecture search for medical image segmentation

b. Nas-fpn: Learning scalable feature pyramid architecture for object detection

c. Dynamic Proposals for Efficient Object Detection

d. DNAS: Decoupling Neural Architecture Search for High-Resolution Remote Sensing Image Semantic Segmentation

2. It would be better if the authors could provide computational complexity analysis in the manuscript.

3. For table 6, it would be better if the authors could compare models with the same architectures. Since the models have different task heads, the table's comparison may be unfair.

Author Response

Thank you very much for your painstaking review work. Your important suggestions make us complete our research article further. In our newly revised manuscript, all the changes are highlighted in blue for clarity. In what follows, we will respond to your questions also in blue under the way of the point-by-point response. In addition, we will enclose an identical response attachment in case of a website display error. 

Point 1:  

The idea of using NAS for dynamic heads has been widely investigated in the current works. It would be better if the authors could discuss and mention them in the related work section.

  1. Nas-unet: Neural architecture search for medical image segmentation
  2. Nas-fpn: Learning scalable feature pyramid architecture for object detection
  3. Dynamic Proposals for Efficient Object Detection
  4. DNAS: Decoupling Neural Architecture Search for High-Resolution Remote Sensing Image Semantic Segmentation.

Response 1:

Thank you very much. We have added these references in our newly revised manuscript.

As for the first two references, we have added descriptions about them in detail. Please see the references [11] and [12], check the descriptions about them in the first paragraph in subsection 2.2 in the revised manuscript (see page 4, lines: 154-164).

As for the third reference, “Dynamic Proposals for Efficient Object Detection”, it is mainly related to object detection. By considering that the term of “dynamic proposals” in their work mainly focuses on object candidates, not for architecture adjustment like our work, thus it is only cited, not described in detail. Please see the reference [59] in this revised manuscript.

As for the fourth reference, “DNAS: Decoupling Neural Architecture Search for High-Resolution Remote Sensing Image Semantic Segmentation”, it has been described in the previous manuscript. Please see the reference [49] in this revised manuscript.

Please kindly check references [11], [12], [49], [59] in our newly revised manuscript.

Point 2: 

It would be better if the authors could provide computational complexity analysis in the manuscript.

Response 2:

Thank you very much. Accurate complexity analysis of a deep learning model should follow along with its architecture. Specifically, by taking the phase of forward mapping in deep model as an example, the computational complexity is first determined by the model itself, including the topology of the hierarchical architecture, number of channels, size of convolutional filters, step size of pooling, and so on. In addition, it is also related to the size of the input image. Formally, in general, the computational complexity scales in roughly about O(h*w*p*s), where h*w is the size of the image, p is the number of the total parameters to be learned in convolutions, and s is a scaling factor related to the neural architecture, including the topology of the hierarchical architecture, the step size of pooling, and so on.

Following your valuable suggestion, we have added some explanations about the relationship between the FLOPs and the computational complexity. That is, in general, the number of model parameters and the number of FLOPs is two important factors to evaluate the computation scale in deep models. Particularly, FLOPs can directly reflect the computational complexity of the model, which is related to the size of the input image and the neural architecture itself.

Accordingly, we have rewritten Subsection 4.5 by adding more analysis about computational complexity. Please kindly check them in this revised manuscript (see pages 20 and 21, lines: 568-597).

Point 3:  

For table 6, it would be better if the authors could compare models with the same architectures. Since the models have different task heads, the table's comparison may be unfair.

Response 3:

Thank you very much. We highly agree your opinion that the comparison experiment should be as fair as possible. We will discuss it separately from aspects of backbone and task head, respectively.

For backbone, we chose the same backbone (HRNet) for HRNet+FCN and HRNet+OCR, and chose the same backbone (ResNet-101) for FCN,  PSPNet, DeepLab v3+ and OCRNet. However, the downsample method of UNet and ResNet is  totally different and they also have different output channels in five consecutive stages. That is the reason why we couldn’t use ResNet-101 as the backbone for UNet. Please refer to the table below for the details.

Model

Number of Output channel

Downsample method

ResNet

(64, 256, 512, 1024, 2048)

MaxPooling

UNet

(64, 128, 256, 512, 1024)

Convolution with stride=2

In addition, Transformers and CNNs are the two large different schools of thought in deep learning. We could not fix the same backbone or head for them.

For task head, we have added a new experiment by taking our DyHRnet as the backbone and that of the FCN as the head. This treatment generates a new model, named as “DyHRNet+FCN”. The performance scores are collected into Table 6 in this revised manuscript. Correspondingly, the number of model parameters and the number of FLOPs in the “DyHRNet+FCN” are added in Table 7.

Some analyses are given. Please kindly check the newly added paragraph (i.e., the last paragraph) in subsection 4.4, the newly added Table 6, and Table 7 in our revised manuscript (see pages 19 and 20, lines: 558-567).

Additionally, we also made several modifications under other reviewer’s suggestions as follows:

  1. According to the suggestion given by reviewer #1 and reviewer #2, five new references have been cited, and discussions about them have been included in the manuscript. Please kindly check the references [5], [11], [12], [42], [59].
  2. According to the suggestion given by reviewer #1, we have rewritten Subsection 4.5 by adding more analyses about computational complexity. (in pages 20 and 21, lines: 568-597)
  3. According to the suggestion given by reviewer #1, we have conducted a new experiment. Please kindly check the newly added Table 6 and the analysis in the last paragraph in Subsection 4.4. (in pages 19 and 20, lines: 558-567)
  4. According to the comment given by reviewer #2, we have redrawn Figure 3, added a new paragraph, and rewritten one previous paragraph below Eq. (4) to explain the combination of NAS and channel attention for clarity. Please kindly check the two paragraphs. (in pages 6 and 7, lines: 237-255)
  5. According to the comment given by reviewer #2, we have rewritten the first paragraph in Subsection 3.2 to explain Figure 2 in detail. Please kindly check it. (in page 7, lines: 278-288)
  6. According to the three questions given by reviewer #3, we have added a new Subsection 5.2, which is titled as “The Behavior of the Accelerated Proximal Gradient Algorithm”. In this subsection, accordingly, three new groups of experiments have been conducted and analyzed. Please kindly check them. (in pages 23 and 24, lines: 633-668)
  7. Some other minors have also been revised in this manuscript.

Thank you very much! Please check our revised manuscript!

Reviewer 2 Report

This paper propose a Dynamic High-Resolution Network for semantic segmentation on RS images. The organization and writing of this paper are well to understand. The paper is relatively innovative. The experiments are also sufficient to verify the effectiveness of the proposed method. In addition, I still have  the following minor concerns.

1. The motivation of the combination of NAS and channel attention is not clear and confuse to me. What  problem are you trying to solve?

2、Three connection networks in Fig.2 show the difference between the proposed method and exsiting similar method. Author should add the experiment comparison of these different connection networks to further verify the effectiveness of the proposed architecture. 

Additionally, some related reference should be add to discuss  in the paper, e.g., 

A Multi-Scale U-Shaped CNN Building Instance Extraction Framework With Edge Constraint for High Spatial Resolution Remote Sensing Imagery, IEEE Transaction on Geosciences and Remote Sensing, 2021

Semantic Segmentation in Aerial Imagery Using Multi-level Contrastive Learning with Local Consistency, WACV, 2023

Author Response

Thank you very much for your thorough review work. Your crucial comments make our revised paper easier to read and comprehend. In our newly revised manuscript, all the changes are highlighted in blue for clarity. In what follows, we will respond to your questions also in blue under the way of the point-by-point response. In addition, we will enclose an identical response attachment in case of a website display error.

Point 1:  

The motivation of the combination of NAS and channel attention is not clear and confuse to me. What problem are you trying to solve?

Response 1:

Thank you very much. There are two components in our proposed model: NAS and channel-wise attention module. The former focuses on the importance of connections between feature maps, and the latter cares about the importance of channels inside the feature maps cube. The combination of them fully realized the dynamic advantages with data adaptability. In this way, we can prune those unimportant connections and channels, and amplify those important connections and channels. We aim to make every parameter of the original HRNet learnable as much as possible. That is also why we call our model dynamic HRNet (DyHRNet). You can also see Eq (4) for a clear explanation, where “s”is learned via the NAS trick, and “a” is evaluated via the channel attention. The comparative results of Table 4 show that both NAS and Channel-wise attention module are effective whenever used alone or in combination.

Following your valuable comment, we have redrawn Figure 3, rewritten the last paragraph in Page 6, and added a new paragraph below Eq (4) to explain the combination of the NAS and channel attention for the development of dynamic HRNet.

Please kindly check the newly added paragraph and the rewritten paragraph below Eq. (4) in this revised manuscript (see pages 6 and 7, lines: 237-255).

Point 2: 

Three connection networks in Fig.2 show the difference between the proposed method and existing similar method. Author should add the experiment comparison of these different connection networks to further verify the effectiveness of the proposed architecture. 

Response 2:  

Thank you very much. We feel very sorry to cause your confusion about it. In the newly revised manuscript version, we have already added detailed describing text in the first paragraph of subsection 3.2. The theme of Figure 2 is not a comparison, but rather describes a process of pruning operation and weighting operation in the order from Fig.2(a) to Fig.2(b) and Fig.2(c).

To be more specific, Fig.2(a) depicts the dense cross-resolution connections of original HRNet, which are also the candidates to be selected by the APG algorithm. At the beginning of training stage, the weights of these connections are all set to be 1.0 to guarantee that all of them have equal probability to be selected. Fig.2(b) visualizes a group of learned weights using solid line and dashed line. The thicker the solid line is, the greater the weight is and the more important the connection is. In particular, the dashed lines indicate those connections are of zero importance, which could be directly cut off. Fig.2(c) shows the finally selected connections, which will be used at the inference stage of DyHRNet.

Naturally, we conducted the comparison experiments between HRNet and DyHRNet (i.e., Fig.2(a) and Fig.2(c).) in Table 1, Table 2, and Table 3.

Point 3:  

Additionally, some related reference should be added to discuss in the paper, e.g., 

A Multi-Scale U-Shaped CNN Building Instance Extraction Framework With Edge Constraint for High Spatial Resolution Remote Sensing Imagery, IEEE Transaction on Geosciences and Remote Sensing, 2021

Semantic Segmentation in Aerial Imagery Using Multi-level Contrastive Learning with Local Consistency, WACV, 2023

Response 3:

Thank you for your kind reminder. The two references above are indeed related to this study. We discuss them in subsection 2.1 in our newly revised manuscript.

Please kindly check the newly added reference [5] and [42], and the descriptions about these works in the second paragraph in subsection 2.1 (see page 3, lines: 129-137).

Additionally, we also made several modifications under other reviewer’s suggestions as follows:

  1. According to the suggestion given by reviewer #1 and reviewer #2, five new references have been cited, and discussions about them have been included in the manuscript. Please kindly check the references [5], [11], [12], [42], [59].
  2. According to the suggestion given by reviewer #1, we have rewritten Subsection 4.5 by adding more analyses about computational complexity. (in pages 20 and 21, lines: 568-597)
  3. According to the suggestion given by reviewer #1, we have conducted a new experiment. Please kindly check the newly added Table 6 and the analysis in the last paragraph in Subsection 4.4. (in pages 19 and 20, lines: 558-567)
  4. According to the comment given by reviewer #2, we have redrawn Figure 3, added a new paragraph, and rewritten one previous paragraph below Eq. (4) to explain the combination of NAS and channel attention for clarity. Please kindly check the two paragraphs. (in pages 6 and 7, lines: 237-255)
  5. According to the comment given by reviewer #2, we have rewritten the first paragraph in Subsection 3.2 to explain Figure 2 in detail. Please kindly check it. (in page 7, lines: 278-288)
  6. According to the three questions given by reviewer #3, we have added a new Subsection 5.2, which is titled as “The Behavior of the Accelerated Proximal Gradient Algorithm”. In this subsection, accordingly, three new groups of experiments have been conducted and analyzed. Please kindly check them. (in pages 23 and 24, lines: 633-668)
  7. Some other minors have also been revised in this manuscript.

Thank you very much! Please check our revised manuscript!

Reviewer 3 Report

I have found this manuscript clear and well written.

Its content provides interesting and original ideas for relevant contributions to the possibility of proposing an architecture learning model for semantic segmentation in the context of neural architecture search (NAS) with a channel-wise attention.

The Authors have provided a well-structured exposition of their material with a gradual description of the underlying ideas.

The content is quite explicit (self-explanatory) and described in sufficient detail to understand the subject, techniques and results.

The experimental part is substantial and offers a clear evaluation of the effectiveness of the proposed method on representative and publicly challenging RS image data sets.

The analysis provided is well conducted and corresponding results are fully appropriate to the text and its content.

The list of references to the literature related to the field is also quite appropriate.

However, I have a few more questions to extend the analysis a little further.

The first comment concerns the initialization scheme chosen. Initially, all weights are set to 1.0 and the procedure is run until convergence, which will lead to a simplification of the final network but will also lead to an unavoidable substantial overhead when learning. I have two questions about this aspect.

1) How does the method (AGP) behave if the weights are initialized in a much more restricted way by adopting a bottom-up approach (very few weights at one and subsequent activation of a certain number of them) rather than a top-down approach (all weights at one and then nullification of a certain number of them)?

2) As it stands, it is difficult to assess the trade-off between the additional cost of calculation and the benefit derived from simplifying the final network; is it possible to quantify or better qualify it?

3) Performing sufficient training remains a key element of the proposed scheme to obtain a relevant update of the weights for the later inference. What is the lower limit, i.e. the percentage of training data below which an outlying value of weights is obtained?

Author Response

Thank you very much for your professional review work. Your insightful comments and inspiring questions enlighten us to explore this study further. In our newly revised manuscript, all the changes are highlighted in blue for clarity. In what follows, we will respond to your questions also in blue under the way of the point-by-point response. In addition, we will enclose an identical response attachment in case of a website display error.

Point 1:

How does the method (APG) behave if the weights are initialized in a much more restricted way by adopting a bottom-up approach (very few weights at one and subsequent activation of a certain number of them) rather than a top-down approach (all weights at one and then nullification of a certain number of them)?

Response 1:

Thank you very much. According to your comments and question, we have conducted three new groups of experiments, including those with different initializations for the weights of the cross-resolution connections, with different ratios of training samples, and with different numbers of iterations. For clarity, we have added a new subsection to collect them together. Please kindly check the newly added subsection 5.2 in this revised manuscript (see pages 23 and 24, lines: 633-668)

As for the comment in this point, we give the explanations as follows:

In Algorithm 1, one of the main tasks is to identify the important cross-resolution connections in the original HRNet for semantic segmentation. To this end, the APG algorithm is employed to solve the sparse regularization subproblem in Problem (6). Algorithmically, it starts with the original HRNet to modulate their weights of the cross-resolution connections, which are all initialized as 1.0 for iterations. This gives equal chance for all the connections to be evaluated. In this way, a top-down training strategy is actually performed, in which all the weights are initialized as 1.0 and then nullified gradually to small scores. After training, the connections with zero weights will be discarded for prediction.

According to your valuable suggestion, we have conducted a new group of experiments with a bottom-up training strategy, where small weight scores are assigned for initialization. To this end, we first assign a small weight to the connections. Specifically, we trained additionally the model with all weights initialized to 0.1 and 0.5, respectively. The performances are collected into the newly added table (see Table 8 in this revised manuscript). In the experiments, it is observed that small initial weights indeed help speed up the convergence, but the performance decreases drastically.

Accordingly, we have added the explanations in this revised manuscript. Please kindly check the first two paragraphs and Table 8 in the newly added subsection 5.2 (see page 23, lines: 634-652).

Point 2:

As it stands, it is difficult to assess the trade-off between the additional cost of calculation and the benefit derived from simplifying the final network; is it possible to quantify or better qualify it?

Response 2:                           

Thank you very much. Yes, it is it is difficult to assess the trade-off between the additional cost of calculation and the benefit derived from simplifying the final network. We explain this point as follows:

Based on the original HRNet, in our work the additional cost of calculation lies in the implementation of the APG algorithm and the optimization of the parameters in the channel-wise attention. However, during training, the additional cost of calculation is controllable. For examples, on the Vaihingen dataset, in the case that the number of iterations is 40000, the total training time is about 12 hours with one NVIDIA GeForce RTX 3090. On the Potsdam and LoveDA datasets, in the case that the number of iterations is 80000, the total training time is about 12 hours. As for the performance enhancement, the scores in Table 1, Table 2, and Table 3 indicate that our DyHRNet achieves significant enhancements, compared with the original HRNet.

In addition, Table 7 lists the number of parameters (#Params) and the number of the FLOPs. They are two important factors to evaluate the computation scale in deep models. From this table, we see that both the #Params and the FLOPs in DyHRNet+FCN are smaller than those in HRNet+FCN. This fact can be also witnessed when HRNet+OCR and DyHRNet+OCR are compared to each other. Thus, it can be concluded that the decrease of the computations in our model largely occurs in the backbone. This is due to the architecture learning, where those connections and channels with zero contributions will be ignored. This indicates the effectiveness of our method.

Please kindly check Tables 1, 2, 3, and 7 in this revised manuscript. Thank you very much.

Point 3:

Performing sufficient training remains a key element of the proposed scheme to obtain a relevant update of the weights for the later inference. What is the lower limit, i.e., the percentage of training data below which an outlying value of weights is obtained?

Response 3:

Thank you very much for your mentioning this point. Yes, performing sufficient training remains a key element of the proposed scheme to obtain a relevant update of the weights for the later inference.

Motivated by your comments and suggestion, we have added two new groups of experiments to investigate the behave of the AGP algorithm. The experiments are conducted with different ratios of training data and different numbers of iterations on the Vaihingen dataset. The goal is to demonstrate whether the weights of the cross-resolution connections learned by the AGP algorithm change drastically.

Specifically, in one group of the newly conducted experiments, the ratios are set as 10%, and 50% of all the total training samples, respectively (Note that the case with all the training samples (100%) had been done in previous manuscript). In another group of the newly conducted experiments, the maximum number of iterations is set as 10000 and 20000, respectively.

Figure 11 visualizes the weights of the cross-resolution connections in the DyHRNet, corresponding to the above four settings respectively. Table 9 reports of the performances of the learned models.  Compared with the model trained on the 100% of the training samples, there are no large changes both in performance scores and in weight values in these settings.

Please kindly check Figure 11, Table 9 and the analyses in the newly added Subsection 5.2 in this revised manuscript (see pages 23 and 24, lines: 653-668)

Additionally, we also made several modifications under other reviewer’s suggestions as follows:

  1. According to the suggestion given by reviewer #1 and reviewer #2, five new references have been cited, and discussions about them have been included in the manuscript. Please kindly check the references [5], [11], [12], [42], [59].
  2. According to the suggestion given by reviewer #1, we have rewritten Subsection 4.5 by adding more analyses about computational complexity. (in pages 20 and 21, lines: 568-597)
  3. According to the suggestion given by reviewer #1, we have conducted a new experiment. Please kindly check the newly added Table 6 and the analysis in the last paragraph in Subsection 4.4. (in pages 19 and 20, lines: 558-567)
  4. According to the comment given by reviewer #2, we have redrawn Figure 3, added a new paragraph, and rewritten one previous paragraph below Eq. (4) to explain the combination of NAS and channel attention for clarity. Please kindly check the two paragraphs. (in pages 6 and 7, lines: 237-255)
  5. According to the comment given by reviewer #2, we have rewritten the first paragraph in Subsection 3.2 to explain Figure 2 in detail. Please kindly check it. (in page 7, lines: 278-288)
  6. According to the three questions given by reviewer #3, we have added a new Subsection 5.2, which is titled as “The Behavior of the Accelerated Proximal Gradient Algorithm”. In this subsection, accordingly, three new groups of experiments have been conducted and analyzed. Please kindly check them. (in pages 23 and 24, lines: 633-668)
  7. Some other minors have also been revised in this manuscript.

Thank you very much! Please check our revised manuscript!

Round 2

Reviewer 1 Report

This is a manuscript after revision. I am grateful for the authors' responses, which have addressed all my concerns.